# GENERATIVE MARGINALIZATION MODELS

## ABSTRACT

We introduce *marginalization models* (MAMs), a new family of generative models for high-dimensional discrete data. They offer scalable and flexible generative modeling with tractable likelihoods by explicitly modeling all induced marginal distributions. Marginalization models enable fast evaluation of arbitrary marginal probabilities with a single forward pass of the neural network, which overcomes a major limitation of methods with exact marginal inference, such as autoregressive models (ARMs). We propose scalable methods for learning the marginals, grounded in the concept of "*marginalization self-consistency*". Unlike previous methods, MAMs also support scalable training of any-order generative models for high-dimensional problems under the setting of *energy-based training*, where the goal is to match the learned distribution to a given desired probability (specified by an unnormalized (log) probability function such as energy or reward function). We demonstrate the effectiveness of the proposed model on a variety of discrete data distributions, including binary images, language, physical systems, and molecules, for *maximum likelihood* and *energy-based training* settings. MAMs achieve orders of magnitude speedup in evaluating the marginal probabilities on both settings. For energy-based training tasks, MAMs enable any-order generative modeling of high-dimensional problems beyond the capability of previous methods.

## 1 INTRODUCTION

Deep generative models have enabled remarkable progress across diverse fields, including image generation, audio synthesis, natural language modeling, and scientific discovery. However, there remains a pressing need to better support efficient probabilistic inference for key questions involving marginal probabilities $p(\mathbf{x}_s)$ and conditional probabilities $p(\mathbf{x}_u|\mathbf{x}_v)$, for appropriate subsets $s, u, v$ of the variables. The ability to directly address such quantities is critical in applications such as outlier detection [50, 40], masked language modeling [11, 72], image inpainting [73], and constrained protein/molecule design [69, 55]. Furthermore, the capacity to conduct such inferences for arbitrary subsets of variables empowers users to leverage the model according to their specific needs and preferences. For instance, in protein design, scientists may want to manually guide the generation of a protein from a user-defined substructure under a particular path over the relevant variables. This requires the generative model to perform arbitrary marginal inferences.

Towards this end, neural autoregressive models (ARMs) [3, 30] have been developed to facilitate conditional/marginal inference based on the idea of modeling a high-dimensional joint distribution as a factorization of univariate conditionals using the chain rule of probability. Many efforts have been made to scale up ARMs and enable any-order generative modeling under the setting of maximum likelihood estimation (MLE) [30, 66, 20], and great progress has been made in applications such as masked language modeling [72] and image inpainting [20]. However, marginal likelihood evaluation in the most widely-used modern neural network architectures (e.g., Transformers [68] and U-Nets [53]) is limited by $\mathcal{O}(D)$ neural network passes, where $D$ is the length of the sequence. This scaling makes it difficult to evaluate likelihoods on long sequences arising in data such as natural language and proteins. In contrast to MLE, in the setting of *energy-based training* (EB), instead of empirical data samples, we only have access to an unnormalized (log) probability function (specified by a reward or energy function) that can be evaluated pointwise for the generative model to match. In such settings, ARMs are limited to fixed-order generative modeling and lack scalability in training. The subsampling techniques developed to scale the training of conditionals for MLE are no longer applicable when matching log probabilities in energy-based training (see Section 4.3 for details).

[?] [C] [=C] [C] [=C] [C] [=C] [Ring1] [=Branch1] ⋯     [Cl] [C] [=C] [C] [=C] [C] [=C] [Ring1] [=Branch1] ⋯     [F] [C] [=C] [C] [=C] [C] [=C] [Ring1] [=Branch1] ⋯

$$p_\theta\left(\bighexagon_{[?]}\right) \; \mathbf{|} \;\; = \;\; p_\theta\left(\bighexagon_{Cl}\right) \, \mathbf{.} \;\; + \;\; p_\theta\left(\bighexagon_{F}\right) \, \mathbf{.}$$

variable $x_1$ is marginalized out

$$p_\theta(?010??) \; \mathbf{.} \quad = \quad p_\theta(0010??) \, \mathbf{.} \quad + \quad p_\theta(1010??) \, \mathbf{.}$$

**Figure 1:** Marginalization models (MAMs) enable estimation of any marginal probability with a neural network $\theta$ that learns to "marginalize out" variables. The figure illustrates marginalization of a single variable on bit strings (representing molecules) with two alternatives (versus $K$ in general) for clarity. The bars represent probability masses.

To enhance scalability and flexibility in the generative modeling of discrete data, we propose a new family of generative models, **marginalization models** (MAMs), that directly model the marginal distribution $p(\mathbf{x}_s)$ for any subset of variables $\mathbf{x}_s$ in $\mathbf{x}$. Direct access to marginals has two important advantages: 1) *significantly speeding up inference for any marginal*, and 2) *enabling scalable training of any-order generative models under both MLE and EB settings*.

The unique structure of the model allows it to simultaneously represent the coupled collection of all marginal distributions of a given discrete joint probability mass function. For the model to be valid, it must be consistent with the sum rule of probability, a condition we refer to as "*marginalization self-consistency*" (see Figure 1); learning to enforce this with scalable training objectives is one of the key contributions of this work.

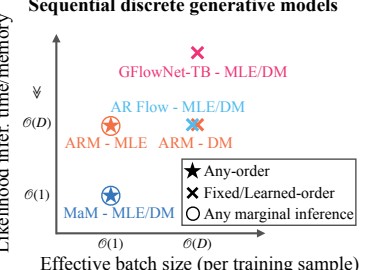

**Figure 2:** Scalability of sequential discrete generative models. The y-axis unit is # of NN forward passes required.

We show that MAMs can be trained under both maximum likelihood and energy-based training settings with scalable learning objectives. We demonstrate the effectiveness of MAMs in both settings on a variety of discrete data distributions, including binary images, text, physical systems, and molecules. We empirically show that MAMs achieve orders of magnitude speedup in marginal likelihood evaluation. For energy-based training, MAMs are able to scale training of any-order generative models to high-dimensional problems that previous methods fail to achieve.

## 2 BACKGROUND

We first review two prevalent generative modeling settings. Then we introduce autoregressive models under two training settings.

**Maximum likelihood (MLE)** Given a dataset $\mathcal{D} = \{\mathbf{x}^{(i)}\}_{i=1}^N$ drawn from a data distribution $p = p_{\text{data}}$, we aim to learn the distribution $p_\theta(\mathbf{x})$ that maximizes the probability of the data under our model. Mathematically, we aim to learn the parameters $\theta^\star$ that maximize the log-likelihood:

$$\theta^\star = \arg\max_\theta \; \mathbb{E}_{\mathbf{x} \sim p_{\text{data}}} \left[ \log p_\theta(\mathbf{x}) \right] \approx \arg\max_\theta \; {}^1\!/_N \sum_{i=1}^N \log p_\theta(\mathbf{x}^{(i)}) \tag{1}$$

which is also equivalent to minimizing the Kullback-Leibler divergence under the empirical distribution, i.e., minimizing $D_{\text{KL}}(p_{\text{data}}(\mathbf{x}) \| p_\theta(\mathbf{x}))$. This is the setting that is most commonly used in generation of images (e.g., diffusion models [59, 18, 60]) and language (e.g. GPT [49]) where we can empirically draw observed data from the distribution.

**Energy-based training (EB)** In this setting, we do not have data from the distribution of interest. Instead, we have access to the unnormalized (log) probability mass function $f$, usually in the form of reward function or energy function, that are defined by humans or by physical systems to specify how likely a sample is. Mathematically, we can define the target probability mass function to be $f(\mathbf{x}) = \exp(r(x)/\tau)$, where $r(x)$ is the reward function and $\tau > 0$ is a temperature parameter. This expresses the intuitive idea that we would like the model to assign higher probability to data with larger reward. For example, the reward function can represent human preferences in alignment of large language models [43, 42]. In molecular/material design applications, scientists can specify the reward according to how close a particular sample's measured or calculated properties are to

some functional desiderata. When modeling the thermodynamic ensemble of physical systems, $r(x)$ is defined to be the (negative) energy function of a given state [41]. Mathematically, we aim to learn the parameters $\theta$ such that $p_\theta(\mathbf{x}) \approx f(\mathbf{x})/Z$, where $Z$ is the normalization constant of $f$. A common training criteria is to minimize the KL divergence [41, 71, 9]:

$$\min_\theta \; D_{\mathrm{KL}}\left(p_\theta\left(\mathbf{x}\right) \,\big\|\, f(\mathbf{x})/Z\right) = \mathbb{E}_{\mathbf{x} \sim p_\theta(\mathbf{x})}\left[\log p_\theta\left(\mathbf{x}\right) - \log f(\mathbf{x})/Z\right] . \tag{2}$$

**Autoregressive models** Autoregressive models (ARMs) [3, 30] model a complex high-dimensional distribution $p(\mathbf{x})$ by factorizing it into univariate conditionals using the chain rule:

$$\log p(\mathbf{x}) = \sum_{d=1}^{D} \log p\left(x_d \mid \mathbf{x}_{<d}\right), \tag{3}$$

where $\mathbf{x}_{<d} = \{x_1, \ldots, x_{d-1}\}$. Recently there has been great success in applying autoregressive models to discrete data, such as natural language, proteins [58, 32, 36], and molecules [56, 15]. Due to their sequential nature via modeling the conditionals, evaluation of (joint/marginal) likelihood requires up to $D$ neural network evaluations. This is costly for long sequences, leading to limitations that prevent ARMs to be scalable for marginal inference and energy-based training.

**Any-order ARMs (AO-ARMs)** Under the MLE setting, Uria et al. [66] propose to learn the conditionals of ARMs for arbitrary orderings that include all permutations of $\{1, \ldots, D\}$. The model $\phi$ can be trained by maximizing a lower-bound objective [66, 20] that takes an expectation under a uniform distribution on orderings. This objective allows scalable training of AO-ARMs, leveraging efficient parallel evaluation of multiple one-step conditionals for each token in one forward pass with architectures such as the U-Net [53] and Transformers [68]. However, under the EB setting, training AO-ARMs presents challenges, which we will discuss in details in Section 4.3.

## 3 MARGINALIZATION MODELS

We propose *marginalization models* (MAMs), a new type of generative model that enables scalable any-order generative modeling as well as efficient marginal evaluation, for both maximum likelihood and energy-based training. The flexibility and scalability of marginalization models are enabled by the explicit modeling of the marginal distribution and enforcing *marginalization self-consistency*.

In this paper, we focus on generative modeling of discrete structures using vectors of discrete variables. The vector representation encompasses various real-world problems with discrete structures, including language sequence modeling, protein design, and molecules with string-based representations (e.g., SMILES [70] and SELFIES [29]). Moreover, vector representations are inherently applicable to any discrete problem, since it is feasible to encode any discrete object into a vector of discrete variables.

**Definition** We are interested in modeling the discrete probability distribution $p(\mathbf{x})$, where $\mathbf{x} = [x_1, \ldots, x_D]$ is a $D$-dimensional vector and each $x_d$ takes $K$ possible values, i.e. $x_d \in \{1, \ldots, K\}$.

**Marginalization** Let $\mathbf{x}_s$ be a subset of variables of $\mathbf{x}$ and $\mathbf{x}_{s^c}$ be the complement set, i.e. $\mathbf{x}_s \subseteq \{x_1, \ldots, x_D\}$ and $\mathbf{x}_{s^c} = \{x_1, \ldots, x_D\} \setminus \mathbf{x}_s$. The marginal of $\mathbf{x}_s$ is obtained by summing over all values of $\mathbf{x}_{s^c}$:

$$p(\mathbf{x}_s) = \sum_{\mathbf{x}_{s^c}} p(\mathbf{x}_s, \mathbf{x}_{s^c}) \tag{4}$$

We refer to (4) as the "*marginalization self-consistency*" that any valid distribution should follow. The goal of a marginalization model $\theta$ is to estimate the marginals $p(\mathbf{x}_s)$ for any subset of variables $\mathbf{x}_s$ as closely as possible. To achieve this, we train a deep neural network $p_\theta$ that minimizes the distance of $p_\theta(\mathbf{x})$ and $p(\mathbf{x})$ on the full joint distribution [1] while enforcing the marginalization self-consistency.

**Parameterization** To approximate arbitrary marginals over $\mathbf{x}_s$ with a single neural network forward pass, we additionally include the "marginalized out" variables $\mathbf{x}_{s^c}$ in the input by introducing a special symbol "?" to denote the missing values. By doing this, we create an augmented $D$-dimensional vector representation $\mathbf{x}_s^{\mathrm{aug}} \in \mathcal{X}^{\mathrm{aug}} \triangleq \{1, \ldots, K, ?\}^D$ and feed it to the NN. For example, for a binary vector $\mathbf{x}$ of length 4, for $\mathbf{x}_s = \{x_1, x_3\}$ with $x_1 = 0$ and $x_3 = 1$, $\mathbf{x}_s^{\mathrm{aug}} = [0, ?, 1, ?]$ where "?" denotes $x_2$ and $x_4$ being marginalized out. From here onwards we will use $\mathbf{x}_s^{\mathrm{aug}}$ and $\mathbf{x}_s$ interchangeably.

---

[1]An alternative is to consider minimizing distance over some marginal distribution of interest if we only cares about a specific marginal. Note this is impractical under the energy-based training setting, when the true marginal $p(\mathbf{x}_s)$ is intractable to evaluate in general.

A marginalization model parameterized by a neural network $\theta$ takes in the augmented vector representation $\mathbf{x}_s^{\text{aug}} \in \{1, \ldots, K, ?\}^D$, and outputs the marginal log probability $f_\theta(\mathbf{x}_s) = \log p_\theta(\mathbf{x}_s)$ that satisfy the marginalization self-consistency constraints[2]:

$$\sum_{\mathbf{x}_{s^c}} p_\theta([\mathbf{x}_s, \mathbf{x}_{s^c}]) = p_\theta(\mathbf{x}_s) \quad \forall \mathbf{x}_s \in \{1, \ldots, K, ?\}^D$$

where $[\mathbf{x}_s, \mathbf{x}_{s^c}]$ denotes the concatenation of $\mathbf{x}_s$ and $\mathbf{x}_{s^c}$. Given a random ordering of the variables $\sigma \in S_D$ where $S_D$ defines the set of all permutations of $1, 2, \cdots, D$, let $\sigma(d)$ denote the $d$-th element in $\sigma$ and $\sigma(< d)$ be the first $d - 1$ elements in $\sigma$. The marginalization can be imposed over one variable at a time, which leads to the following one-step marginalization constraints:

$$p_\theta(\mathbf{x}_{\sigma(<d)}) = \sum_{x_{\sigma(d)}} p_\theta(\mathbf{x}_{\sigma(\leq d)}), \quad \forall \sigma \in S_D, \mathbf{x} \in \{1, \cdots, K\}^D, d \in [1 : D]. \tag{5}$$

**Sampling**  Given the learned marginalization model, one can sample from the learned distribution by picking an arbitrary order $\sigma$ and sampling one variable at a time. To evaluate the conditionals at each step of the generation, we can use the product rule of probability:

$$p_\theta(x_{\sigma(d)}|\mathbf{x}_{\sigma(<d)}) = p_\theta(\mathbf{x}_{\sigma(\leq d)}) \, / \, p_\theta(\mathbf{x}_{\sigma(<d)}).$$

However, the above is not a valid conditional distribution if the marginalization in (5) is not strictly enforced, since it might not sum up exactly to one. Hence we use following normalized conditional:

$$p_\theta(x_{\sigma(d)}|\mathbf{x}_{\sigma(<d)}) = \frac{p_\theta([\mathbf{x}_{\sigma(<d)}, x_{\sigma(d)}])}{\sum_{x_{\sigma(d)}} p_\theta([\mathbf{x}_{\sigma(<d)}, x_{\sigma(d)}])}. \tag{6}$$

In this paper, we focus on the sampling procedure that generates one variable at a time, but marginalization models can also facilitate sampling multiple variables at a time ( See Appendix B.2 ).

**Scalable learning of marginals with conditionals**  In training, we impose the marginalization self-consistency by minimizing the *squared error* of the constraints in (5) in log-space. Evaluation of each marginalization constraint in (5) requires $K$ NN forward passes, where $K$ is the number of discrete values $x_d$ can take. This makes training challenging to scale when $K$ is large. To address this issue, we augment the marginalization models with learnable conditionals parameterized by $\phi$. The marginalization constraints in (5) can be decomposed into $K$ parallel marginalization constraints, which makes it *highly scalable* to subsample from for training:

$$p_\theta(\mathbf{x}_{\sigma(<d)})p_\phi(\mathbf{x}_{\sigma(d)}|\mathbf{x}_{\sigma(<d)}) = p_\theta(\mathbf{x}_{\sigma(\leq d)}), \qquad \forall \sigma \in S_D, \mathbf{x} \in \{1, \cdots, K\}^D, d \in [1 : D]. \tag{7}$$

During training, we need to specify a distribution $q(\mathbf{x})$ for subsampling the marginalization constraints to optimize on. In practice, it can be set to the distribution we are interested to perform marginal inference on, such as $p_{\text{data}}$ or the distribution of the generative model $p_{\theta,\phi}$.

## 4 TRAINING THE MARGINALIZATION MODELS

### 4.1 MAXIMUM LIKELIHOOD ESTIMATION TRAINING

In this setting, we train MAMs with the maximum likelihood objective while additionally enforcing the marginalization constraints in Equation (5):

$$\max_{\theta,\phi} \quad \mathbb{E}_{\mathbf{x} \sim p_{\text{data}}} \log p_\theta(\mathbf{x}) \tag{8}$$

$$\text{s.t.} \quad p_\theta(\mathbf{x}_{\sigma(<d)})p_\phi(\mathbf{x}_{\sigma(d)}|\mathbf{x}_{\sigma(<d)}) = p_\theta(\mathbf{x}_{\sigma(\leq d)}), \, \forall \sigma \in S_D, \mathbf{x} \in \{1, \cdots, K\}^D, d \in [1 : D].$$

**Two-stage training**  A typical way to solve the above optimization problem is to convert the constraints into a penalty term and optimize the penalized objective, but we empirically found the learning to be slow and unstable. Instead, we identify an alternative two-stage optimization formulation that is theoretically equivalent to Equation (8), but leads to more efficient training:

**Claim 1.** *Solving the optimization problem in* (8) *is equivalent to the following two-stage optimization procedure, under mild assumption about the neural networks used being universal approximators:*

***Stage 1:***  $\max_\phi \ \mathbb{E}_{\mathbf{x} \sim p_{data}}\mathbb{E}_{\sigma \sim \mathcal{U}(S_D)} \sum_{d=1}^{D} \log p_\phi \left( x_{\sigma(d)} \mid \mathbf{x}_{\sigma(<d)} \right)$

***Stage 2:***  $\min_\theta \ \mathbb{E}_{\mathbf{x} \sim q(\mathbf{x})}\mathbb{E}_{\sigma \sim \mathcal{U}(S_D)}\mathbb{E}_{d \sim \mathcal{U}(1,\cdots,D)} \left( \log[p_\theta(\mathbf{x}_{\sigma(<d)})p_\phi(\mathbf{x}_{\sigma(d)}|\mathbf{x}_{\sigma(<d)})] - \log p_\theta(\mathbf{x}_{\sigma(\leq d)}) \right)^2.$

---

[2]To make sure $p_\theta$ is normalized, we can either additionally enforce $p_\theta([? ? \cdots ?]) = 1$ or let $Z_\theta = p_\theta([? ? \cdots ?])$ be the normalization constant.

The first stage can be interpreted as *fitting the conditionals* in the same way as AO-ARMs [66, 20] and the second stage acts as *distilling the marginals* from conditionals. The intuition comes from the chain rule of probability: there is a one-to-one correspondence between optimal conditionals $\phi$ and marginals $\theta$, i.e. $\log p_\theta(\mathbf{x}) = \sum_{d=1}^{D} \log p_\phi \big( x_{\sigma(d)} | \mathbf{x}_{\sigma(<d)} \big)$ for any $\sigma$ and $\mathbf{x}$. By assuming neural networks are universal approximators, we can first optimize for the optimal conditionals, and then optimize for the corresponding optimal marginals. We provide more details in Appendix A.1.

## 4.2 Energy-based Training

In this setting, we train MAMs using the energy-based training objective in Equation (2) with a penalty term to enforce the marginalization constraints in Equation (5):

$$\min_{\theta,\phi} D_{\mathrm{KL}}\big(p_\theta(\mathbf{x}) \,\|\, p(\mathbf{x})\big) + \lambda\, \mathbb{E}_{\mathbf{x}\sim q(\mathbf{x})} \mathbb{E}_\sigma \mathbb{E}_d \big(\log\big[p_\theta\big(\mathbf{x}_{\sigma(<d)}\big)\, p_\phi\big(\mathbf{x}_{\sigma(d)}|\mathbf{x}_{\sigma(<d)}\big)\big] - \log p_\theta\big(\mathbf{x}_{\sigma(\leq d)}\big)\big)^2,$$

where $\sigma \sim \mathcal{U}(S_D)$, $d \sim \mathcal{U}(1, \cdots, D)$ and $q(\mathbf{x})$ is the distribution of interest for evaluating marginals.

**Scalable training**   We use REINFORCE to estimate the gradient of the KL divergence term:

$$\nabla_\theta D_{\mathrm{KL}}(p_\theta(\mathbf{x})\|p(\mathbf{x})) = \mathbb{E}_{\mathbf{x}\sim p_\theta(\mathbf{x})} \left[ \nabla_\theta \log p_\theta(\mathbf{x}) \left(\log p_\theta(\mathbf{x}) - \log f(\mathbf{x})\right) \right]$$
$$\approx {}^{1}\!/\!_N \sum_{i=1}^{N} \nabla_\theta \log p_\theta(\mathbf{x}^{(i)}) \left(\log p_\theta(\mathbf{x}^{(i)}) - \log f(\mathbf{x}^{(i)})\right) \qquad (9)$$

For the penalty term, we subsample the ordering $\sigma$ and step $d$ for each data $\mathbf{x}$.

**Efficient sampling with persistent MCMC**   We need cheap and effective samples from $p_\theta$ in order to perform REINFORCE, so a persistent set of Markov chains are maintained by randomly picking an ordering and taking block Gibbs sampling steps using the conditional distribution $p_\phi(\mathbf{x}_{\sigma(d)}|\mathbf{x}_{\sigma(<d)})$ (full algorithm in Appendix A.5), in similar fashion to persistent contrastive divergence [64]. The samples from the conditional distribution $p_\phi$ serve as approximate samples from $p_\theta$ when they are close to each other. Otherwise, we can additionally use importance sampling for adjustment.

## 4.3 Addressing limitations of ARMs

We discuss in more detail about how MAMs address some limitations of ARMs. The first one is general to both training settings, while the latter two are specific to energy-based training.

1) **Slow marginal inference of likelihoods**   Due to sequential conditional modeling, evaluation of a marginal $p_\phi(\mathbf{x}_o)$ with ARMs (or an arbitrary marginal with AO-ARMs) requires applying the NN $\phi$ up to $D$ times, which is inefficient in time and memory for high-dimensional data. In comparison, MAMs are able to estimate any arbitrary marginal with one NN forward pass.

2) **Lack of support for any-order training**   In energy-based training, the objective in Equation (2) aims to minimize the distance between $\log p_\phi(\mathbf{x})$ and $\log p(x)$, where $\phi$ is the NN parameters of an ARM. However, unless the ARM is perfectly self-consistent over all orderings, it will not be the case that $\log p_\phi(\mathbf{x}) = \mathbb{E}_\sigma \log p_\phi(\mathbf{x}|\sigma)$. Therefore, the expected $D_{\mathrm{KL}}$ objective over the orderings $\sigma$ would not be equivalent to the original $D_{\mathrm{KL}}$ objective, i.e., $\mathbb{E}_{p_\phi}[\mathbb{E}_\sigma \log p_\phi(x \mid \sigma) - \log p(x)] \neq \mathbb{E}_{p_\phi}[\log p_\phi(x) - \log p(x)]$. As a result, ARMs cannot be trained with the expected $D_{\mathrm{KL}}$ objective over all orderings simultaneously, but instead need to resort to a preset order and minimize the KL divergence between $\log p_\phi(\mathbf{x}|\sigma)$ and the target density $\log p(\mathbf{x})$. The self-consistency constraints imposed by MAMs address this issue. MAMs are not limited to fixed ordering because marginals are order-agnostic and we can optimize over expectation of orderings for the marginalization self-consistency constraints.

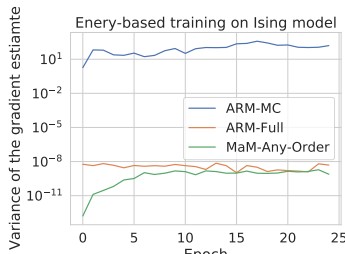

**Figure 3:** Approximating $\log p_\phi(\mathbf{x})$ with one-step conditional (ARM-MC) results in extremely high gradient variance during energy-based training.

3) **Training not scalable on high-dimensional problems**   When minimizing the difference between $\log p_\phi(\mathbf{x}|\sigma)$ and the target $\log p(\mathbf{x})$, ARMs need to sum conditionals to evaluate $\log p_\phi(\mathbf{x}|\sigma)$. One might consider subsampling one-step conditionals $p_\phi(x_{\sigma(d)}|\mathbf{x}_{\sigma(<d)})$ to estimate $p_\phi(\mathbf{x})$, but this leads

to high variance of the REINFORCE gradient in Equation (9) due to the product of the score function and distance terms, which are both high variance (We validate this in experiments, see Figure 3). Consequently, training ARMs for energy-based training necessitates a sequence of $D$ conditional evaluations to compute the gradient of the objective function. This constraint leads to an effective batch size of $B \times D$ for batch of $B$ samples, significantly limiting the scalability of ARMs to high-dimensional problems. Furthermore, obtaining Monte Carlo samples from ARMs for the REINFORCE gradient estimator is slow when the dimension is high. Due to the fixed input ordering, this process requires $D$ sequential sampling steps, making more cost-effective sampling approaches like persistent MCMC infeasible. Marginalization models circumvent this challenge by directly estimating the log-likelihood with the marginal neural network. Additionally, the support for any-order training enables efficient sampling through the utilization of persistent MCMC methods.

## 5 RELATED WORK

**Autoregressive models**  Developments in deep learning have greatly advanced the performance of ARMs across different modalities, including images, audio, and text. Any-order (Order-agnostic) ARMs were first introduced in [66] by training with the any-order lower-bound objective for the maximum likelihood setting and recently seen in ARDM [20] with state-of-the-art performance for any-order discrete modeling of image/text/audio. Germain et al. [16] train an auto-encoder with masking that outputs the sequence of all one-step conditionals for a given ordering, but does not generate as well as methods [67, 72, 20] that predict one-step conditionals under the given masking. Douglas et al. [14] trains an AO-ARM and use importance sampling to estimate arbitrary conditional posteriors, but with limited experiment validation on a synthetic dataset. Shih et al. [57] utilizes a modified training objective of ARMs for better marginal inference performance but loses any-order generation capability. Comparisons of MAMs and ARMs are discussed in detail in Section 4.3.

**Arbitrary conditional/marginal models**  For continuous data, VAEAC [25] and ACFlow [31] extends the idea of conditional variational encoder and normalizing flow to model arbitrary conditionals. ACE [62] improves the expressiveness of arbitrary conditional models through directly modeling the energy function, which puts less constraints on parameterization but comes at the cost of approximating the normalizing constant. Instead of using neural networks as function approximators, probabilistic circuits (PCs) [6, 45] offer tractable probabilistic models for both conditionals and marginals by building a computation graph with sum and product operations following specific structural constraints. Examples of PCs include Chow-Liu trees [7], arithmetic circuits [10], sum-product networks [47], etc. Peharz et al. [45] have improved the scalability of PCs through combining arithmetic operations into a single monolithic einsum-operation and automatic differentiation. More recently, [33, 34] demonstrated the potential of PCs with distilling latent variables from trained deep generative models on continuous image data. However, their expressiveness are limited by the structural constraints. All methods mentioned above focus on MLE settings, except ARMs are explored in energy-based training of science problems [9, 71], but suffer in scaling when $D$ is large.

**GFlowNets**  GFlowNets [2, 4] formulate the problem of generation as matching the probability flow at terminal states to the target normalized density. Compared to ARMs, GFlowNets allow flexible modeling of the generation process by assuming learnable generation paths through a directed acyclic graph (DAG). The advantages of learnable generation paths come with the trade-off of sacrificing the flexibility of any-order generation and exact likelihood evaluation. Under fixed generation path, GFlowNets are reduced to fixed-order ARMs [74]. In Appendix A.3, we further identify the connections and differences between GFlowNets and AO-ARMs/MAMs. For discrete problems, Zhang et al. [75] train GFlowNets on the squared distance loss with the trajectory balance objective [38], which is less scalable for large $D$ (due to the same reason as ARMs in Section 4.3) and renders direct access to marginals unavailable. For the MLE setting, an energy function is additionally learned from data such that training is reduced to energy-based training.

## 6 EXPERIMENTS

We conduct experiments with marginalization models (MAM) on both MLE and EB settings for discrete problems including binary images, text, molecules and phyiscal systems. We consider the following baselines for comparison: Any-order ARM (AO-ARM) [20], ARM [30], GFlowNet [39, 75], Discrete Flow[65] and Probabilistic Circuit (PC) [45]. MAM, PC and

Original    Censored-100    Censored-400    Censored-700    Generated-100    Generated-400    Generated-700

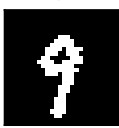 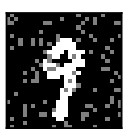 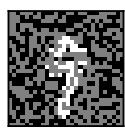 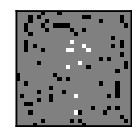 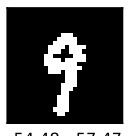 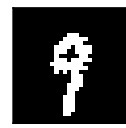 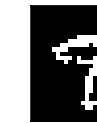

-54.48, -57.47     -60.48, -63.37     -106.45, -108.58

**Figure 4:** An example of the data generated (with 100/400/700 pixels masked) for comparing the quality of likelihood estimate. Numbers below the images are LL estimates from MAM's marginal network (left) and AO-ARM-E's ensemble estimate (right).

**Table 1:** Performance Comparison on Binary-MNIST

| Model | NLL (bpd) ↓ | Spearman's ↑ | Pearson ↑ | Marg. inf. time (s) ↓ |
|---|---|---|---|---|
| AO-ARM-E-U-Net | **0.148** | **1.0** | **1.0** | 661.98 ± 0.49 |
| AO-ARM-S-U-Net | 0.149 | 0.996 | 0.993 | 132.40 ± 0.03 |
| GflowNet-MLP | 0.189 | – | – | – |
| PC-Image (EiNets)[4] | 0.187 | 0.716 | 0.752 | **0.015 ± 0.00** |
| MAM-U-Net | 0.149 | 0.992 | 0.993 | 0.018 ± 0.00 |

(AO-)ARM support arbitrary marginal inference. Discrete flow[3] allows exact likelihood evaluation while GFlowNet needs to approximate the likelihood with sum using importance samples. For evaluating AO-ARM's marginal inference, we can either use an ensemble model by averaging over several random orderings (AO-ARM-E) or use a single random ordering (AO-ARM-S). In general, AO-ARM-E should always be better than AO-ARM-S but at a much higher cost. Neural network architecture and training hyperparameter details can be found in Appendix C.

Ablation studies on measuring marginal self-consistency and sampling with marginals are in Appendices B.1 and B.2.

Guidance on picking $q$ is in Appendix B.3. Appendix C.3 contains more results on CIFAR-10.

### 6.1 MAXIMUM LIKELIHOOD ESTIMATION TRAINING

**Binary MNIST** We report the negative test likelihood (bits/digit), marginal estimate quality and marginal inference time per minibatch (of size 16) in Table (1). To keep GPU memory usage the same, we sequentially evaluate the likelihood for ARMs. Both MAM and AO-ARM use a U-Net architecture with 4 ResNet Blocks interleaved with attention layers (see Appendix C). GFlowNets fail to scale to large architectures as U-Net, hence we report GFlowNet results using an MLP from Zhang et al. [75]. For MAM, we use the conditional network to evaluate test likelihood (since this is also how MAM generates data). The marginal network is used for evaluating marginal inference. The quality of the marginal estimates will be compared to the best performing model.

In order to evaluate the quality of marginal likelihood estimates, we employ a controlled experiment where we randomly mask out portions of a test image and generate multiple samples with varying levels of masking (refer to Figure 4). This process allows us to obtain a set of distinct yet comparable samples, each associated with a different likelihood value. For each model, we evaluate the likelihood of the generated samples and compare that with AO-ARM-E's estimate since it achieves the best likelihood on test data. We repeat this controlled experiment on a random set of test images. The mean Spearman's and Pearson correlation are reported to measure the strength of correlation in marginal inference likelihoods between the given model and AO-ARM-E. MAM achieves close to 4 *order of magnitude speed-up* in marginal inference while at *comparable quality* to that from AO-ARM-S. PCs are also very fast in marginal inference but there remains a gap in terms of quality. Generated samples and additional marginal inference on partial images are in Appendix C.

**Molecular sets (MOSES)** We test generative modeling of MAM on a benchmarking molecular dataset [46] refined from the ZINC database [61]. Same metrics are reported as Binary-MNIST. Likelihood quality is measured similarly but on random groups of test molecules instead of generated ones. The generated molecules from MAM and AO-ARM are comparable to standard state-of-the-art molecular generative models, such as CharRNN [56], JTN-VAE [26], and LatentGAN [48] (see Appendix C), with additional controllability and flexibility in any-order generation. MAM supports

---

[3]Results are only reported on text8 for discrete flow since there is no public code implementation.

[4]We adopt the SOTA implementation of PCs from EiNets [45]. Results are reported on Binary MNIST using the image-tailored PC structure [47]. For text and molecular data, designing tailored PC structures that deliver competitive performance remains an open challenge.

**Table 2:** Performance Comparison on Molecular Sets

| Model | NLL (bpd) ↓ | Spearman's ↑ | Pearson ↑ | Marg. inf. time (s) ↓ |
|---|---|---|---|---|
| AO-ARM-E-Transfomer | **0.652** | **1.0** | **1.0** | 96.87± 0.04 |
| AO-ARM-S-Transformer | 0.655 | 0.996 | 0.994 | 19.32± 0.01 |
| MAM-Transfomer | 0.655 | 0.998 | 0.995 | **0.006±0.00** |

**Table 3:** Performance Comparison on text8

| Model | NLL (bpc) ↓ | Spearman's ↑ | Pearson ↑ | Marg. inf. time (s) ↓ |
|---|---|---|---|---|
| Discrete Flow (8 flows) | 1.23 | – | – | – |
| AO-ARM-E-Transformer | **1.494** | **1.0** | **1.0** | 207.60 ± 0.33 |
| AO-ARM-S-Transformer | 1.529 | 0.982 | 0.987 | 41.40 ± 0.01 |
| MAM-Transformer | 1.529 | 0.937 | 0.945 | **0.005 ± 0.000** |

**Table 4:** Performance Comparison on Ising model ($10 \times 10$)

| Model | NLL (bpd) ↓ | KL divergence ↓ | Marg. inf. time (s) ↓ |
|---|---|---|---|
| ARM-Forward-Order-MLP | 0.79 | **-78.63** | 5.29±0.07e-01 |
| ARM-MC-Forward-Order-MLP | 24.84 | -18.01 | 5.30±0.07e-01 |
| GFlowNet-Learned-Order-MLP | **0.78** | -78.17 | – |
| MAM-Any-Order-MLP | 0.80 | -77.77 | **3.75±0.08e-04** |

**Table 5:** Performance Comparison on Target Lipophilicity

| Model | KL divergence ↓ | | | |
|---|---|---|---|---|
| **Distribution** | logP $= 4, \tau = 1.0$ | logP $= -4, \tau = 1.0$ | logP $= 4, \tau = 0.1$ | logP $= 4, \tau = 0.1$ |
| ARM-FO-MLP | -174.25 | -168.62 | -167.83 | -160.2 |
| MAM-AO-MLP | -173.07 | -166.43 | -165.75 | -157.59 |

much faster marginal inference, which is useful for domain scientists to reason about likelihood of (sub)structures. Generated molecules and property histogram plots of are available in Appendix C.

**Text8** Text8 [37] is a widely used character level natural language modeling dataset. The dataset comprises of 100M characters from Wikipedia, split into chunks of 250 character. We follow the same testing procedure as Binary-MNIST and report the same metrics. The test NLL of discrete flow is from [65], for which there are no open-source implementations to evaluate additional metrics.

## 6.2 ENERGY-BASED TRAINING

We compare with ARM that uses sum of conditionals to evaluate $\log p_\phi$ with fixed forward ordering and ARM-MC that uses a one-step conditional to estimate $\log p_\phi$. ARM can be regarded as the golden standard of learning autoregressive conditionals, since its gradient needs to be evaluated on the full generation trajectory, which is the most informative and costly. MAM uses marginal network to evaluate $\log p_\theta$ and subsamples a one-step marginalization constraint for each data point in the batch. The effective batch size for ARM and GFlowNet is $B \times \mathcal{O}(D)$ for batch of size $B$, and $B \times \mathcal{O}(1)$ for ARM-MC and MAM . MAM and ARM optimizes KL divergence using REINFORCE gradient estimator with baseline. GFlowNet is trained on per-sample gradient of squared distance [75].

**Ising model** Ising models [24] model interacting spins and are widely studied in mathematics and physics (see MacKay [35]). We study Ising model on a square lattice. The spins of the $D$ sites are represented a $D$-dimensional binary vector and its distribution is $p^*(\mathbf{x}) \propto f^*(\mathbf{x}) = \exp\left(-\mathcal{E}_J(\mathbf{x})\right)$ where $\mathcal{E}_\mathbf{J}(\mathbf{x}) \triangleq -\mathbf{x}^\top \mathbf{J}\mathbf{x} - \boldsymbol{\theta}^\top \mathbf{x}$, with $\mathbf{J}$ the binary adjacency matrix. These models, although simplistic, bear analogies to the complex behavior of high-entropy alloys [9]. We compare MAM with ARM, ARM-MC, and GFlowNet on a $10 \times 10$ ($D=100$) and a larger $30 \times 30$ ($D=900$) Ising model where ARMs and GFlowNets fail to scale. 2000 ground truth samples are generated following Grathwohl et al. [17] and we measure test negative log-likelihood on those samples. We also measure $D_{\mathrm{KL}}(p_\theta(\mathbf{x})||p^*)$ by sampling from the learned model and evaluating $\sum_{i=1}^{M}(\log p_\theta(\mathbf{x}_i) - \log f^*(\mathbf{x}_i))$. Figure 5 contains KDE plots of $-\mathcal{E}_\mathbf{J}(\mathbf{x})$ for the generated samples. As described in Section 4.3, the ARM-MC gradient suffers from high variance and fails to converge. It also tends to collapse and converge to a single sample. MAM has significant speedup in marginal inference and is the only model that supports any-order generative modeling. The performance in terms of KL divergence and likelihood are only slightly worse than models with fixed/learned order, which is expected since any-order modeling is harder than fixed-order modeling, and MAM is solving a more complicated task

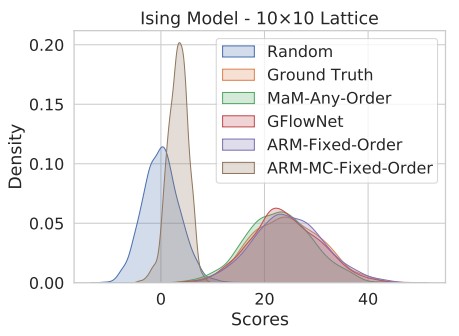 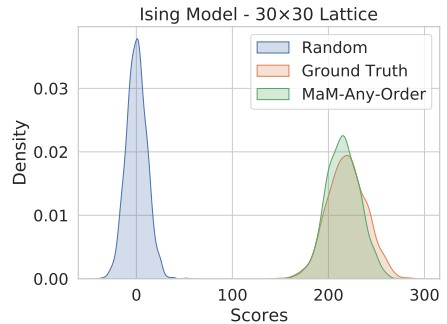

**Figure 5:** Ising model: 2000 samples are generated for each method.

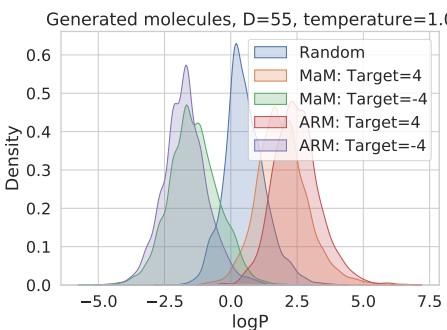 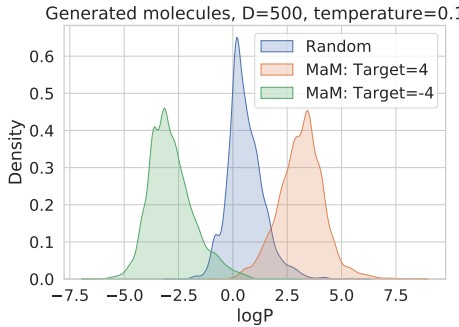

**Figure 6:** Target property matching: 2000 samples are generated for each method.

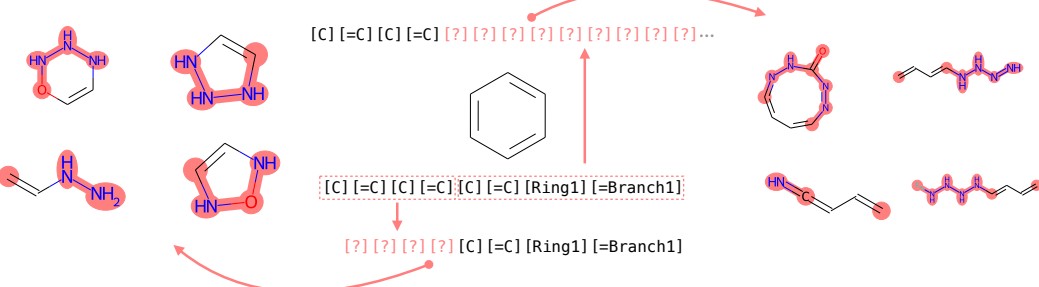

**Figure 7:** Conditionally generate towards low lipophilicity from a user-defined substructure in any given order. Left: Masking out the left 4 SELFIES characters. Right: Masking the right 4-20 SELFIES characters.

of jointly learning conditionals and marginals. On a $30 \times 30$ ($D = 900$) Ising model, MAM achieves a bpd of $0.835$ on ground-truth samples while ARM and GFlowNet fails to scale. Distribution of generated samples is shown in Figure 5.

**Molecular generation with target property** In this task, we are interested in training generative models towards a specific target property of interest $g(x)$, such as lipophilicity (logP), synthetic accessibility (SA) etc. We define the distribution of molecules to follow $p^*(x) \propto \exp(-(g(x) - g^*)^2/\tau)$, where $g^*$ is the target value of the property and $\tau$ is a temperature parameter. We train ARM and MAM for lipophilicity of target values $4.0$ and $-4.0$, both with $\tau = 1.0$ and $\tau = 0.1$. Both models are trained for 4000 iterations with batch size 512. Results are shown in Figure 6 and Table 5 (additional figures in Appendix C). Findings are consistent with the Ising model experiments. Again, MAM performs just marginally below ARM. However, only MAM supports any-order modeling and scales to high-dimensional problems. Figure 6 (right) shows molecular generation with MAM for $D = 500$.

## 7    CONCLUSION

In conclusion, marginalization models are a novel family of generative models for high-dimensional discrete data that offer scalable and flexible generative modeling with tractable likelihoods. These models explicitly model all induced marginal distributions, allowing for fast evaluation of arbitrary marginal probabilities with a single forward pass of the neural network. MAMs also support scalable training objectives for any-order generative modeling, which previous methods struggle to achieve under the energy-based training setting. Potential future work includes designing new neural network architectures that automatically satisfy the marginalization self-consistency.

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

# A    ADDITIONAL TECHNICAL DETAILS

## A.1    PROOF OF PROPOSITION 1

*Proof.* From the single-step marginalization self-consistency in (7), we have

$$\log p_\theta(\mathbf{x}) = \sum_{d=1}^{D} \log p_\phi\left(x_{\sigma(d)}|\mathbf{x}_{\sigma(<d)}\right), \ \forall \mathbf{x}, \sigma.$$

Therefore we can rewrite the optimization in (8) as:

$$\max_{\phi} \quad \mathbb{E}_{\mathbf{x} \sim p_{\text{data}}} \mathbb{E}_{\sigma \sim \mathcal{U}(S_D)} \sum_{d=1}^{D} \log p_\phi\left(x_{\sigma(d)} \mid \mathbf{x}_{\sigma(<d)}\right) \tag{10}$$

$$\text{s.t.} \quad p_\theta(\mathbf{x}_{\sigma(<d)})p_\phi(\mathbf{x}_{\sigma(d)}|\mathbf{x}_{\sigma(<d)}) = p_\theta(\mathbf{x}_{\sigma(\le d)}), \ \forall \sigma \in S_D, \mathbf{x} \in \{1, \cdots, K\}^D, d \in [1:D].$$

Let $p^*$ be the optimal probability distribution that maximizes the likelihood on training data, and from the chain rule we have:

$$p^* = \arg\max_{p} \mathbb{E}_{\mathbf{x} \sim p_{\text{data}}} \log p(\mathbf{x}) = \mathbb{E}_{\mathbf{x} \sim p_{\text{data}}} \mathbb{E}_{\sigma \sim \mathcal{U}(S_D)} \sum_{d=1}^{D} \log p\left(x_{\sigma(d)}|\mathbf{x}_{\sigma(<d)}\right)$$

Then $p^*$ is also the optimal solution to (10) the marginalization constraints are automatically satisfied by $p^*$ since it is a valid distribution. From the universal approximation theorem [23, 22, 8], we can use separate neural networks to model $p_\theta$ (marginals) and $p_\phi$ (conditionals), and obtain optimal solution to (10) with $\theta^*$ and $\phi^*$ that approximates $p^*$ arbitrarily well.

Specifically, if $\theta^*$ and $\phi^*$ satisfy the following three conditions below, they are the optimal solution to (10):

$$p_{\phi^*}\left(x_{\sigma(d)} \mid \mathbf{x}_{\sigma(<d)}\right) = p^*\left(x_{\sigma(d)} \mid \mathbf{x}_{\sigma(<d)}\right), \quad \forall \mathbf{x}, \sigma \tag{11}$$

$$p_{\theta^*}(\mathbf{x}_s) = p^*(\mathbf{x}_s)Z_{\theta^*}, \quad \forall \mathbf{x}, s \subseteq \{1, \cdots, D\} \tag{12}$$

$$p_{\theta^*}(\mathbf{x}_{\sigma(<d)})p_{\phi^*}(\mathbf{x}_{\sigma(d)}|\mathbf{x}_{\sigma(<d)}) = p_{\theta^*}(\mathbf{x}_{\sigma(\le d)}), \ \forall \sigma \in S_D, \mathbf{x} \in \{1, \cdots, K\}^D, d \in [1:D] \tag{13}$$

where $Z_{\theta^*}$ is the normalization constant of $p_{\theta^*}$ and is equal to $p_{\theta^*}((?, \cdots, ?))$. It is easy to see from the definition of conditional probabilities that satisfying any two of the optimal conditions leads to the third one.

To obtain the optimal $\phi^*$, it suffices to solve the following optimization problem:

$$\textbf{Stage 1:} \quad \max_{\phi} \mathbb{E}_{\mathbf{x} \sim p_{\text{data}}} \mathbb{E}_{\sigma \sim \mathcal{U}(S_D)} \sum_{d=1}^{D} \log p_\phi\left(x_{\sigma(d)} \mid \mathbf{x}_{\sigma(<d)}\right)$$

because $p^* = \arg\max_p \mathbb{E}_{\mathbf{x} \sim p_{\text{data}}} \mathbb{E}_{\sigma \sim \mathcal{U}(S_D)} \sum_{d=1}^{D} \log p^*\left(x_{\sigma(d)}|\mathbf{x}_{\sigma(<d)}\right)$ due to chain rule. Solving Stage 1 is equivalent to finding $\phi^*$ that satisfies condition (11). Then we can obtain the optimal $\theta^*$ by solving for condition (13) given the optimal conditionals $\phi^*$:

$$\textbf{Stage 2:} \quad \min_{\theta} \mathbb{E}_{\mathbf{x} \sim q(\mathbf{x})} \mathbb{E}_{\sigma \sim \mathcal{U}(S_D)} \mathbb{E}_{d \sim \mathcal{U}(1, \cdots, D)} \left(\log[p_\theta(\mathbf{x}_{\sigma(<d)})p_{\phi^*}(\mathbf{x}_{\sigma(d)}|\mathbf{x}_{\sigma(<d)})] - \log p_\theta(\mathbf{x}_{\sigma(\le d)})\right)^2$$

$$\square$$

## A.2    EXPECTED LOWER BOUND OF LOG-LIKELIHOOD

Here we present the expected lower bound objective used for training AO-ARMs under maximum likelihood setting, which was first proposed by Uria et al. [66]. Hoogeboom et al. [20] provided the expected lower bound perspective.

Given an ordering $\sigma$,

$$\log p(\mathbf{x} \mid \sigma) = \sum_{d=1}^{D} \log p\left(x_{\sigma(d)} \mid \mathbf{x}_{\sigma(<d)}\right). \tag{14}$$

By taking the expectation over all orderings $\sigma$, we can derive a lower bound on the log-likelihood via Jensen's inequality.

$$\log p_\phi(\mathbf{x}) = \log \mathbb{E}_\sigma p_\phi(\mathbf{x} \mid \sigma) \overset{\text{Jensen's inequality}}{\ge} \mathbb{E}_\sigma \sum_{d=1}^{D} \log p_\phi\left(x_{\sigma(d)} \mid \mathbf{x}_{\sigma(<d)}\right)$$

$$= \mathbb{E}_{\sigma \sim \mathcal{U}(S_D)} D \, \mathbb{E}_{d \sim \mathcal{U}(1,...,D)} \log p_\phi\left(x_{\sigma(d)} \mid \mathbf{x}_{\sigma(<d)}\right)$$

$$= D \, \mathbb{E}_d \, \mathbb{E}_\sigma \frac{1}{D-d+1} \sum_{j \in \sigma(\ge d)} \log p_\phi\left(x_j \mid \mathbf{x}_{\sigma(<d)}\right), \tag{15}$$

where $\sigma \sim \mathcal{U}(S_D)$, $d \sim \mathcal{U}(1,\ldots,D)$ and $\mathbf{x}_{\sigma(<d)} = \{x_{\sigma(1)},\ldots,x_{\sigma(d-1)}\}$. $\mathcal{U}(S)$ denotes the uniform distribution over a finite set $S$ and $\sigma(d)$ denotes the $d$-th element in the ordering.

## A.3 Connections between MaMs and GFlowNets

In this section, we identify an interesting connection between generative marginalization models and GFlowNets. The two type of models are designed with different motivations. GFlowNets are motivated by learning a policy to generate according to an energy function and MaMs are motivated from any-order generation through learning to perform marginalization. However, under certain conditions, there exists an interesting connection between generative marginalization models and GFlowNets. In particular, the marginalization self-consistency condition derived from the definition of marginals in Equation (4) has an equivalence to the "detailed balance" constraint in GFlowNet under the following specific conditions.

**Observation 1.** *When the directed acyclic graph (DAG) used for generation in GFlowNet is specified by the following conditions, there is an equivalence between the marginalization self-consistency condition in Equation (7) for MAM and the detailed balance constraint proposed for GFlowNet [4]. In particular, the $p_\theta(x_{\sigma(d)}|\mathbf{x}_{\sigma(<d)})$ in MAM is equivalent to the forward policy $P_F(\mathbf{s}_{d+1} \mid \mathbf{s}_d)$ in GFlowNet, and the marginals $p_\theta(x_{\sigma(d)})$ are equal to the flows $F(\mathbf{s}_d)$ up to a normalizing constant.*

- *DAG Condition: The DAG used for generation in GFlowNet is defined by the given tree-like structure: a sequence $\mathbf{x}$ is generated by incrementally adding one variable at each step, following a uniformly random ordering $\sigma$ i.e. $\sigma \sim \mathcal{U}(S_D)$. At step $d$, the state along the generation trajectory is defined to be $\mathbf{s}_d = \mathbf{x}_{\sigma(\leq d)}$.*

- *Backward Policy Condition: At step $D - d$, the backward policy under the DAG is fixed by removing (un-assigning) the value of the $d + 1$-th element under ordering $\sigma$ , i.e. $P_B(\mathbf{s}_d \mid \mathbf{s}_{d+1}; \sigma) = \mathbb{1}_{\{\mathbf{s}_d = \mathbf{x}_{\sigma(\leq d)}\}}$. Or equivalently, the backward policy removes (un-assigns) one of the existing variables at random, i.e. $P_B(\mathbf{s}_d \mid \mathbf{s}_{d+1}) = 1/d+1 \mathbb{1}_{\{\mathbf{s}_d \subset \mathbf{s}_{d+1}\}}$.*

Intuitively, this is straight forward to understand, since GFlowNet generates a discrete object autoregressively. The model was proposed to enhance the flexibility of generative modeling by allowing for a learned ordering, as compared with auto-regressive models (see [75] Sec. 5 for a discussion). When the generation ordering is fixed, it is reduced to autoregressive models with fixed ordering, which is discussed in [74]. Observation 1 presented above for any-order ARMs can be seen as a extended result of the connection between GFlowNets and fixed-order ARMs.

We have seen the interesting connection of GFlowNets with ARMs (and MaMs). Next, we discuss the differences between GFlowNets and MaMs.

**Remark 1.** *The detailed balance constraint was proposed only as a theoretical result in Bengio et al. [4]. In actual experiments, GFlowNets are trained using either flow matching [2] or trajectory balance [38, 75].*

Zhang et al. [75] is the most relevant GFlowNet work that targets the discrete problem setting. Training is done via minimizing the squared distance loss with trajectory balance objective. For the MLE training, it proposes to additionally learn an energy function from data so that the trajectory balance objetive can still be applied. In particular, MAM is different from GFlowNet in Zhang et al. [75] in three main aspects.

- First of all, MaMs target any-order generation and direct access to marginals, where as GFlowNets aim for flexibility in learning generation paths and does not offer exact likelihood or direct access to marginals under learnable generation paths. When the generation path is fixed to follow a ordering or random ordering, they are reduced to ARMs or any-order ARMs, which allow for exact likelihood. However, training with the trajectory balance objective does not offer direct access to marginals (just like how ARMs do not offer direct access to marginals but only conditionals).

- Second, training under MLE setting is signiticantly different: GFlowNets learn an additional learned energy function to reduce MLE training back to energy-based training, while MaMs directly maximizes the expected lower bound on the log-likelihood under the marginalization self-consistent constraint.

- Lastly, the training objective is different under energy-based training. GFlowNets are trained on squared distance under the expectation to be specified to be either on-policy, off-policy, or a mixture of both. MAMs are trained on KL divergence where the expectation is defined to be on-policy. It is possible though to train MAMs with squared distance and recently Malkin et al. [39] have shown the equivalence of the gradient of KL divergence and the on-policy expectation of the per-sample gradient of squared distance (which is the gradient actually used for training GFlowNets).

## A.4 DISCUSSION OF NEURAL GENERATIVE MODELS AND PROBABILISTIC CIRCUITS

Probabilistic circuits [6, 45, 33, 34] are very powerful and promising approaches that exhibit great properties such as fast and exact marginalization through smart design of the model's structure and operations.

On the other hand, neural generative models perform approximate inference utilizing the powerful expressiveness of neural networks. Compared to PCs, the neural approximate inference approaches do not have the exact marginalization properties but have better flexibility in modeling complex distributions [57, 20]. Hence there is a trade-off between exact marginalization v.s. approximate marginalization. Our work falls under the neural generative models category but directly approximates marginals. Direct modeling of marginals opens opportunites for more flexible sampling, as shown in Appendix B.2, and more scalable approximate marginal inference and training under EB settings.

## A.5 ALGORITHMS

We present the algorithms for training MAM for maximum likelihood and energy-based training settings in Algorithm 1 and Algorithm 2.

---

**Algorithm 1** MLE training of MAMs

**Input**: Data $\mathcal{D}_{\text{train}}$, $q(\mathbf{x})$, network $\theta$ and $\phi$
**Stage 1:** Train $\phi$ with Equation (15) used in AO-ARM
**for** minibatch $\mathbf{x} \sim \mathcal{D}_{\text{train}}$ **do**
    Sample $\sigma \sim \mathcal{U}(S_D)$, $d \sim \mathcal{U}(1, \cdots, D)$
    $\mathcal{L} \leftarrow \frac{D}{D-d+1} \sum_{j \in \sigma(\geq d)} \log p_\phi \left( x_j | \mathbf{x}_{\sigma(<d)} \right)$
    Update $\phi$ with gradient of $\mathcal{L}$
**end for**
**Stage 2:** Train $\theta$ to distill the marginals from optimized conditionals $\phi$
**for** minibatch $\mathbf{x} \sim q(\mathbf{x})$ **do**
    Sample $\sigma \sim \mathcal{U}(S_D)$, $d \sim \mathcal{U}(1, \cdots, D)$
    $\mathcal{L} \leftarrow$ squared error of the inconsistencies in Equation (7)
    Update $\theta$ with gradient of $\mathcal{L}$
**end for**

---

**Algorithm 2** Energy-based training of MAMs

**Input**: $q(\mathbf{x})$, network $\theta$ and $\phi$, Gibbs sampling block size $M$
**Joint training of $\phi$ and $\theta$:**
**for** $j$ in $\{1, \cdots, N\}$ **do**
    Sample $\sigma \sim \mathcal{U}(S_D)$
    Update $\mathbf{x} \sim p_\phi(\mathbf{x}_{\sigma(\leq M)}|\mathbf{x}_{\sigma(>M)})$
             ▷ Persistent block Gibbs sampling
    Sample $\tilde{\mathbf{x}} \sim q(\mathbf{x})$
    Sample $\tilde{d} \sim \mathcal{U}(1, \cdots, D)$, $\tilde{\sigma} \sim \mathcal{U}(S_D)$
    $\mathcal{L}_{\text{penalty}} \leftarrow$ squared error of Equation (7), for $\tilde{d}$ and $\tilde{\sigma}$ with $\tilde{\mathbf{x}}$
    $\nabla_{\theta,\phi} D_{\text{KL}} \leftarrow$ REINFORCE est. with $\mathbf{x}$
    $\nabla_{\theta,\phi} \leftarrow \nabla_{\theta,\phi} D_{\text{KL}} + \lambda \nabla_{\theta,\phi} \mathcal{L}_{\text{penalty}}$
    Update $\theta$ and $\phi$ with gradient
**end for**

---

## A.6 ADDITIONAL LITERATURE ON DISCRETE GENERATIVE MODELS

**Discrete diffusion models** Discrete diffusion models learn to denoise from a latent base distribution into the data distribution. Sohl-Dickstein et al. [59] first proposed diffusion for binary data and was extended in Hoogeboom et al. [21] for categorical data and both works adds uniform noise in the diffusion process. A wider range of transition distributions was proposed in Austin et al. [1] and insert-and-delete diffusion processes have been explored in Johnson et al. [27]. Hoogeboom et al. [20] explored the connection between ARMs and diffusion models with absorbing diffusion and showed that OA-ARDMs are equivalent to absorbing diffusion models in infinite time limit, but achieves better performance with a smaller number of steps.

**Discrete normalizing flow** Normalizing flows transform a latent base distribution into the data distribution by applying a sequence of invertible transformations [52, 63, 12, 59, 51, 13, 28, 44]. They have been extended to discrete data [65, 19] with carefully designed discrete variable transformations. Their performance is competitive on character-level text modeling, but they do not allow any-order modeling and could be limited to discrete data with small number of categories due to the use of a straight-through gradient estimators.

# B ABLATION STUDIES

## B.1 SCRUTINIZING MARGINAL SELF-CONSISTENCY

The *marginal self-consistency* in MAMs is enforced through optimizing the scalable training objective. Here we empirically examine how well they are enforced in practice. First we look at *checkerboard*, a synthetic problem often used for testing clustering algorithms. More recently it has been used for testing and visualizing both continuous and discrete generative models. We define a discrete input space by discretizing the continuous coordinates of points in 2D. To be more concrete, the origin range $[-4, 4]$ of each dimension is converted into a 16-bit string following the standard way of converting float to string. The target unnormalized probability $p(\mathbf{x})$ is set to 1 for points within dark squares and $1e - 10$ within light squares (since it is infeasible to set it to $\ln(0) = -\infty$ for a NN to learn, and in practice $1e - 10$ is negligible compared to 1). We trained a 5-layer MLP with hidden node size 2048 and residual connections on this problem on both MLE and EBM settings and $q(\mathbf{x})$ is set to be a balanced mixture of ground truth data and samples from $p_\theta$ for MLE or uniform random for EBM:

$$\min_\theta -\mathbb{E}_{\mathbf{x} \sim p_{\text{data}}} p_\theta(\mathbf{x}) + \lambda \, \mathbb{E}_{\mathbf{x} \sim q(\mathbf{x})} \mathbb{E}_\sigma \mathbb{E}_d \Big( \log \sum_{x_{\sigma(d)}} p_\theta([\mathbf{x}_{\sigma(<d)}, x_{\sigma(d)}]) - \log p_\theta([\mathbf{x}_{\sigma(<d)}, x_{\sigma(d)}]) \Big)^2.$$

$$\min_\theta D_{\text{KL}}(p_\theta(\mathbf{x}) \| p(\mathbf{x})) + \lambda \, \mathbb{E}_{\mathbf{x} \sim q(\mathbf{x})} \mathbb{E}_\sigma \mathbb{E}_d \Big( \log \sum_{x_{\sigma(d)}} p_\theta([\mathbf{x}_{\sigma(<d)}, x_{\sigma(d)}]) - \log p_\theta([\mathbf{x}_{\sigma(<d)}, x_{\sigma(d)}]) \Big)^2.$$

For this problem, only a marginal network $\theta$ is trained to predict the $\log p$ of any marginals. Upon training to convergence, the generative models perform on par or better than state of the art discrete generative models and achieve a 20.68 test NLL. See Figure 10 for a comparison of ground truth and learned PMF heatmap. It can be seen the PMF are approximated quite accurately. We investigate how well the marginal self-consistency are enforced, by looking at the marginal estimates of MAMs trained with $\lambda = 1e2$ and $\lambda = 1e4$. We evaluate marginals over the first dimension ($0 - 16$ bits) by fixing the second dimension ($17 - 32$ bits) to 1.0 (bit string $= 0001111111111111$). We plot marginals by marginalizing out bit $3 - 16$ (i.e. $(x_1, x_2, \cdots)$) and bit $5 - 16$ (i.e. $(x_1, x_2, x_3, x_4, \cdots)$). In Figure 11, when $\lambda = 1e4$, the self-consistency are more strictly enforced, leading to matched marginals. Notice that there is some tiny residue PMF at the light squares with the $1e - 10$ approximation applied to points with 0 probability, but they are negligible compared to the significant probability masses. After normalizing the marginals over all possibilities, the marginals are almost exactly matched. In Figure 12, when $\lambda = 1e2$, the self-consistency are more loosely enforced as compared to $\lambda = 1e4$. But it is notable that they are only shifted by a constant as compared to the ground truth marginals. This means although marignal self-consisteny is not strictly enforced when $\lambda = 1e2$, softly enforcing it leads to shifted but consistent estimates of marginals. Using the constant-shifted marginals to sample will result in the same distribution with the ground truth, since after normalization, MAM marginals match the ground truth almost exactly. This is observed in the samples generated under $\lambda = 1e2$ in Figure 10 and consistent normalized marginals in Figure 12. Another interesting observation is that when the marginal self consistency is more softly enforced, less mass is placed on

## B.2 SAMPLING WITH MARGINALS V.S. CONDITIONALS

The trained marginalization model comes with two networks. The conditional network $\phi$ estimates any-order conditionals $p_\phi(\mathbf{x}_{\sigma(d)}|\mathbf{x}_{\sigma(<d)})$, and the marginal network $\theta$ estimates arbitrary marginals $p_\theta(\mathbf{x}_{\sigma(\leq d)})$. When MAM is used for sampling, either network can be used. With the conditional network $\phi$, samples can be drawn autoregressively one variable at each step. Or the marginals can be used to draw variables using the normalized conditional:

$$p_\theta(\mathbf{x}_{s_i}|\mathbf{x}_{s(<i)}) = \frac{p_\theta([\mathbf{x}_{s_i}, \mathbf{x}_{s(<i)}])}{\sum_{\mathbf{x}_{s_i}} p_\theta([\mathbf{x}_{s_i}, \mathbf{x}_{s(<i)}])}.$$

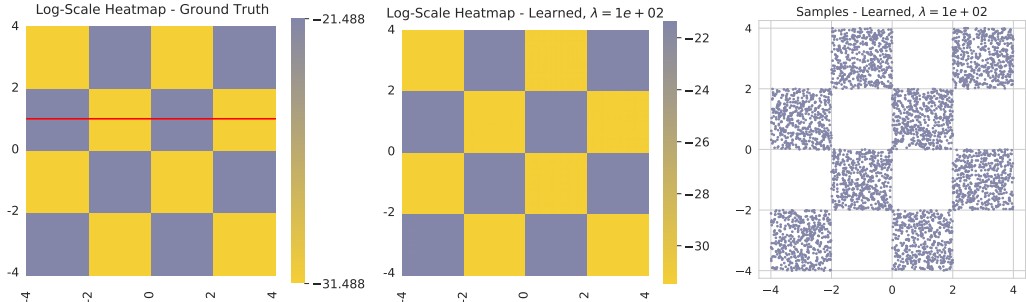

**Figure 8:** PMF heat map under EB training. The learned PMF and ground truth PMF are consistent to each other relatively well. The MSE on $\log p$ (or $p$) of dark pixels is $0.0033$ (or $7.67e - 20$) and the MSE on light pixels is $0.0076$ (or $3.73e - 30$). We are evaluating marginals along the red line: i.e. fixing $(\mathbf{x}_{17}, \cdots, \mathbf{x}_{32}) = (0, 0, 0, 1, 1, 1, \cdots, 1)$, which correspond to $1$ in floating number for y-axis, and perform marginalization over $(\mathbf{x}_1, \cdots, \mathbf{x}_{16})$. $(0, 0, \cdots)$ corresponds to $[0, 2]$. $(0, 1, \cdots)$ corresponds to $[2, 4]$. $(1, 0, \cdots)$ corresponds to $[-2, 0]$. $(1, 1, \cdots)$ corresponds to $[-4, -2]$.

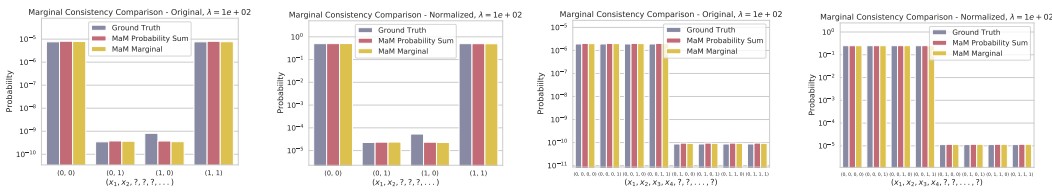

**Figure 9:** Marginal consistency $\lambda = 1e2$ under EB training. Ground truth: summing over ground truth PMF. MAM Probability Sum: summing over learned PMF from MAM. MAM Marginal: direct estimate with MAM. The small discrepancy in $p(1, 0, ?, \cdots, ?)$ is due to the corner case of $(1, 0, 0, 0, \cdots, 0)$ be assigned to a positive value due to numerical errors in float conversion.

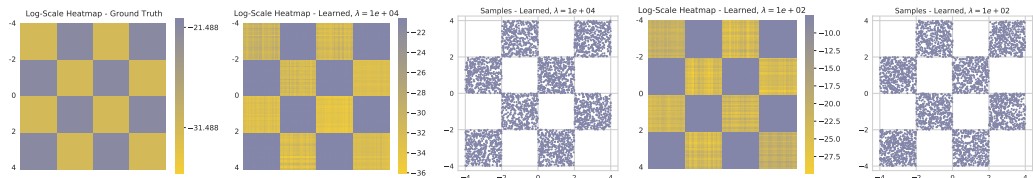

**Figure 10:** PMF heat map under MLE training. The learned PMF and ground truth PMF are consistent to each other relatively well. The MSE on $\log p$ (or $p$) of dark pixels is $0.533$ (or $2.5e - 19$) and the MSE on light pixels is $2.5$ (or $3e - 28$).

where $\mathbf{x}_{s_i}$ is the next block of variables (can be multiple) to sample at step $i$ and $\mathbf{x}_{s(<i)}$ are the previously sampled variables. We show with experiments that the marginals are also effective to be used for sampling and they provide extra flexibility in the sampling procedure. We test sampling with different block sizes using the marginals with random orderings and compare them to sampling with conditionals in Figure 13. The samples generated are of similar quality. And those different sampling procedures exhibit similar likelihood on test data. However, sampling with large block size enables to trade compute memory for less time spent (due to fewer steps) in generation inference, which we find it interesting to explore for future work. Compared with the conditional network, the marginal network allows sampling in arbitrary block variable size and ordering. This illustrates the potential utility of MAMs in flexible generation with tractable likelihood.

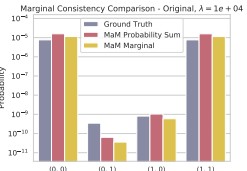 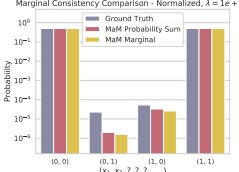 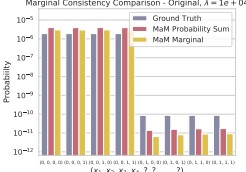 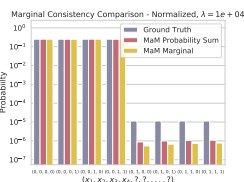

**Figure 11:** Marginal consistency $\lambda = 1e4$ with MLE training. Ground truth: summing over ground truth PMF. MAM Probability Sum: summing over learned PMF from MAM. MAM Marginal: direct estimate with MAM. Note that $p$ for $(0, 1)$ and $(1, 0)$ should be in principle close to zero, but are non-zero due to float-to-int converting numerical errors.

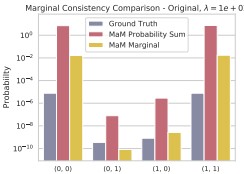 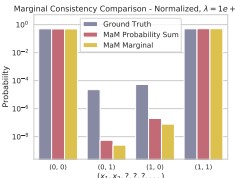 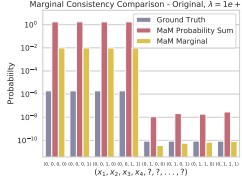 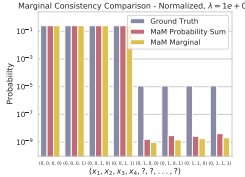

**Figure 12:** Marginal consistency $\lambda = 1e2$ with MLE training.

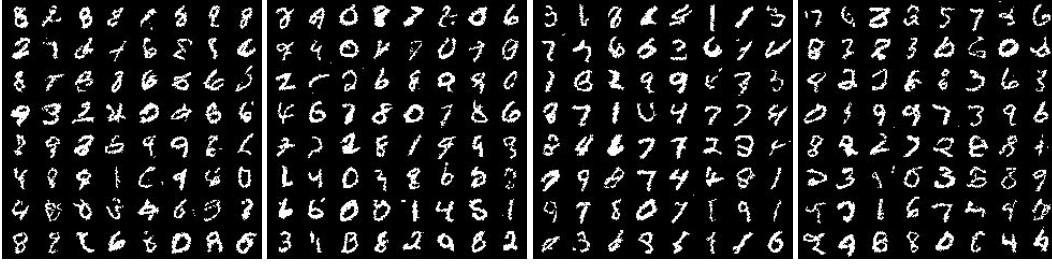

**(a)** Marg.: 1 pixel per step  **(b)** Marg.: 2 pixels per step  **(c)** Marg.: 4 pixels per step  **(d)** Marg.: 8 pixels per step

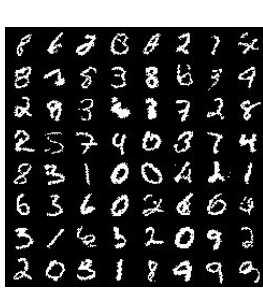
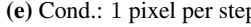

**(e)** Cond.: 1 pixel per step

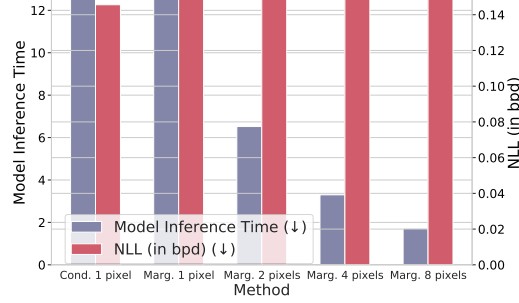

**(f)** Inference time and NLL comparison

**Figure 13:** MAM sampling using marginal network (a-d) with different number of variables at each step v.s. sampling using conditional network (e) with 1 variable at each step. (f) compares NLL under different sampling procedures and the model inference time.

### B.3 CHOICE OF $q$ IN SAMPLING THE MARGINALIZATION SELF-CONSISTENCY FOR TRAINING

In simple examples such as the synthetic checkerboard problem, it does not really matter, we have tried $p_{\text{data}}$ or $p_\theta$ or random, or a mixture of them. All work fairy well given that the problem is relatively easy.

In real-world data problems, it boils down to what the marginal will be used for at test time. Uniform distribution over $x$ will be a bad choice if there is a data manifold we care about. If it will be used for generation, for example in the MNIST Binary example in Appendix B.2, $q$ is set to a mixture of $p_\theta$ and $p_{data}$. If it will be used for mariginal inference on the data manifold, $p_{data}$ will be enough. We all know the NN is not robust on data it hasn't trained on, and so are the marginal networks, they will not give correct estimates if we evaluate on arbitrary datapoint off the manifold or policy.

## C  ADDITIONAL EXPERIMENTS DETAILS

### C.1  DATASET DETAILS

**Binary MNIST**  Binary MNIST is a dataset introduced in [54] that stochastically set each pixel to "1" or "0" in proportion to its pixel intensity. We use the training and test split of [54] provided in https://github.com/yburda/iwae/tree/master [5].

To evaluate the quality of the likelihood estimates, we employ a controlled experiment where we randomly mask out portions ($100$, $400$, and $700$ pixels) of a test image and generate multiple samples with varying levels of masking (refer to Figure 4). We repeat this for $160$ (randomly subsampled) test images and created a dataset of $640$ sets of comparable images. To further test the quality of the marginal likelihood estimates on partially observed images, we curate a dataset of $160$ sets of partial test images ($7 \sim 9$ images in each set) by randomly subsampling from the test set and masking the upper half of the images. To make sure the partial images are comparable but different in their log-likelihood, in each set, we remove samples that have a log-likelihood close to another sample within the threshold of $5.0$.

**Molecular Sets**  The molecules in MOSES are represented either in SMILES [70] or SELFIES [29] strings. We construct a vocabulary (including a stop token) from all molecules and use discrete valued strings to represent molecules. It is worth noting that MAM can also be applied for modeling molecules at a coarse-grained level with predefined blocks, which we leave for future work.

The test set used for evaluating likelihood estimate quality is constructed in a similar manner to Binary MNIST, by drawing sets of random samples from the test dataset.

**text8**  In this dataset, we use a vocabulary of size 27 to represent the letter alphabet with an extra value to represent spaces.

The test set of datasets used for evaluating likelihood estimate quality is constructed in a similar manner to Binary MNIST, each set is generated by randomly masking out portions of a test text sequence (by $50$, $100$, $150$, $200$ tokens) and generating samples.

**Ising model**  The Ising model is defined on a 2D cyclic lattice. The $\mathbf{J}$ matrix is defined to be $\sigma \mathbf{A}_N$, where $\sigma$ is a scalar and $\mathbf{A}_N$ is the adjacency matrix of a $N \times N$ grid. Positive $\sigma$ encourages neighboring sites to have the same spins and negative $\sigma$ encourages them to have opposite spins. The bias term $\boldsymbol{\theta}$ places a bias towards positive or negative spins. In our experiments, we set $\sigma$ to $0.1$ and $\boldsymbol{\theta}$ to $\mathbf{1}$ scaled by $0.2$. Since we only have access to the unnormalized probability, we generate $2000$ samples following [17] using Gibbs sampling with $1,000,000$ steps for $10 \times 10$ and $30 \times 30$ lattice sizes. Those data serve as ground-truth samples from the Ising model for evaluating the test log-likelihood.

**Molecular generation with target property**  During training, we need to optimize on the loss objective on samples generated from the neural network model. However, if the model generates SMILES strings, not all strings correspond to a valid molecule, which makes training at the start challenging when most generated SMILES strings are invalid molecules. Therefore, we use SELFIES string representation as it is a $100\%$ robust in that every SELFIES string corresponds to a valid molecule and every molecule can be represented by SELFIES.

### C.2  TRAINING DETAILS

**Binary MNIST**

- "0", "1" and "?" are represented by a scalar value ("?" takes the value 0) and additionally a mask indicating if it is a "?".

- U-Net with 4 ResNet Blocks interleaved with attention layers for both AO-ARM and MAM. MAM uses two separate neural networks for learning marginals $\phi$ and conditionals $\theta$. Input resolution is $28 \times 28$ with 256 channels used.

- The mask is concatenated to the input. $3/4$ of the channels are used to encode input. The remaining $1/4$ channels encode the mask cardinality (see [20] for details).

- MAM first learns the conditionals $\phi$ and then learns the marginals $\theta$ by finetuning on the downsampling blocks and an additional MLP with 2 hidden layers of dimension 4096. We observe it is necessary to finetune not only on the additional MLP but also on the downsampling blocks to get a good estimate of the marginal probability, which shows marginal network and conditional network rely on different features to make the final prediction.

- Batch size is 128, Adam is used with learning rate 0.0001. Gradient clipping is set to 100. Both AO-ARM and MAM conditionals are trained for 600 epochs. The MAM marginals are finetuned from the trained conditionals for 100 epochs.

**MOSES and text8**

- Transformer with 12 layers, 768 dimensions, 12 heads, 3072 MLP hidden layer dimensions for both AO-ARM and MAM. Two separate networks are used for MAM.

- SMILES or SELFIES string representation and "?" are first converted into one-hot encodings as input to the Transformer.

- MAM first learns the conditionals $\phi$ and then learns the marginals $\theta$ by finetuning on the MLP of the Transformer.

- Batch size is 512 for MOSES and 256 for text8.

- AdamW is used with learning rate 0.0005, betas 0.9/0.99, weight decay 0.001. Gradient clipping is set to 0.25. Both AO-ARM and MAM conditionals are trained for 1000 epochs for text8 and 200 epochs for MOSES. The MAM marginals are finetuned from the trained conditionals for 200 epochs.

**Ising model and molecule generation with target property**

- Ising model input are of $\{0, 1, ?\}$ values and are one-encoded as input to the neural network. The same is done for molecule SELFIES strings.

- MLP with residual layers, 3 hidden layers, feature dimension is 2048 for Ising model. 6 hidden layers, feature dimension 4096 for molecule target generation.

- Adam is used with learning rate of 0.0001. Batch size is 512 and 4096 for molecule target generation. ARM, GFlowNet and MAM are trained with $19,800$ steps for the Ising model. ARM and MAM are trained with $3,000$ steps for molecule target generation.

- Separate networks are used for conditionals and marginals of MAM. They are trained jointly with penalty parameter $\lambda$ set to 4.

**Compute**

- All models are trained on a single NVIDIA A100. The evaluation time is tested on an NVIDIA GTX 1080Ti.

## C.3 ADDITIONAL RESULTS ON CIFAR-10

Due to limited time, we are only able to train MaMs for 180 epochs and achieve a test NLL of 3.34 bpd (if we continue training to 3000 epochs, test NLL will get close to 2.69 bpd shown in the AO-ARM literature [20]). The MaM marginal quality are evaluated using the same procedure on spearman's and pearson correlation with the AO-ARM model, with 0.9796 and 0.9774 respectively. The marginal self-consistency error is averaged $\sim 0.3$ in $\log p$ values. It is conceivable that the same conclusion will apply when training conditionals for more number of epochs and learning marginals from conditionals.

Test NLL is compared in Table 6. MaM achieve highly correlations in terms of $\log p$ estimate when compared with AO-ARM $\log p$'s. Generated samples are shown in Figure 14 and Figure 15. Results are consistent with Hoogeboom et al. [20] and Shih et al. [57].

**Table 6:** Performance Comparison on CIFAR-10

| Model | NLL (bpd) ↓ | Spearman's ↑ | Pearson ↑ | Marg. inf. time (s) ↓ |
|---|---|---|---|---|
| AO-ARM-S-U-Net | **3.344** | - | - | 2400.94 ± 185.72 |
| MAM-U-Net | **3.344** | 0.978 | 0.977 | **0.50 ± 0.02** |

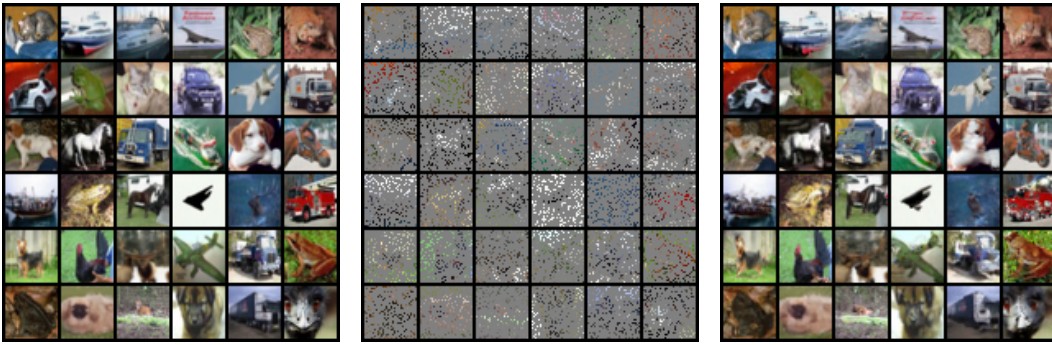

**Figure 14:** . Conditional generation from masked images. *Left*: Original test images. *Middle*: Masked images. *Right*: conditionally generated images.

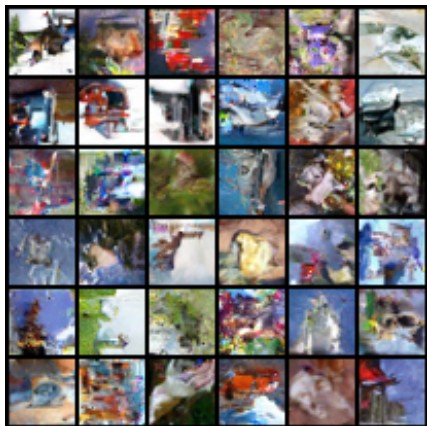

**Figure 15:** . Generated samples. Note that sometimes images are flipped because MaM is trained on augmented images.

## C.4  ADDITIONAL RESULTS ON BINARY MNIST

**Likelihood estimate on partial Binary MNIST images**

Figure 16 illustrates an example set of partial images that we evaluate and compare likelihood estimate from MAM against ARM. Table 7 contains the comparison of the marginal likelihood estimate quality and inference time.

**Generated image samples**

Figure 17 shows how a digit is generated pixel-by-pixel following a random order. We show generated samples from MAM using the learned conditionals $\phi$ in Figure 18.

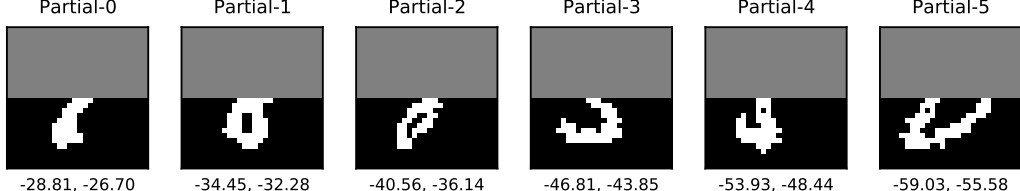

**Figure 16:** An example set of partial images for evaluating marginal likelihood estimate quality. The numbers in the captions show the log-likelihood calculated using learned marginals (left) v.s. learned conditionals (right)

**Table 7:** Performance Comparison on Binary-MNIST partial images

| Model | Spearman's ↑ | Pearson ↑ | Marg. inf. time (s) ↓ |
|---|---|---|---|
| AO-ARM-E-U-Net | **1.0** | **1.0** | $248.96 \pm 0.14$ |
| AO-ARM-S-U-Net | **1.0** | 0.997 | $49.75 \pm 0.03$ |
| MAM-U-Net | 0.998 | 0.995 | $\mathbf{0.02 \pm 0.00}$ |

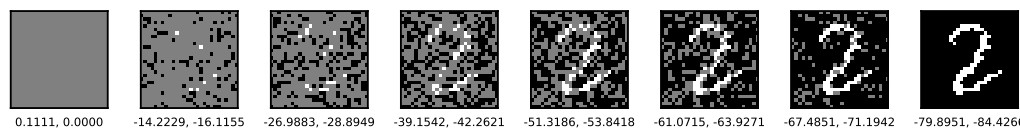

**Figure 17:** An example of the trajectory every 112 step when generating an MNIST digit following a random order. The future pixels are generated by conditioning on the existent filled-in pixels. The numbers in the captions show the log-likelihood calculated using learned marginals (left) v.s. learned conditionals (right)

## C.5 ADDITIONAL RESULTS ON MOSES

**Table 8:** Performance Comparison on MOSES

| Model | Valid↑ | Unique 10k↑ | Frag Test↓ | Scaf TestSF↑ | Int Div1↑ | Int Div2↑ | Filters↑ | Novelty↑ |
|---|---|---|---|---|---|---|---|---|
| Train | 1.0 | 1.0 | 1.0 | 0.9907 | 0.8567 | 0.8508 | 1.0 | 1.0 |
| HMM | 0.076 | 0.5671 | 0.5754 | 0.049 | 0.8466 | 0.8104 | 0.9024 | **0.9994** |
| NGram | 0.2376 | 0.9217 | 0.9846 | 0.0977 | **0.8738** | **0.8644** | 0.9582 | **0.9694** |
| CharRNN | **0.9748** | 0.9994 | **0.9998** | **0.1101** | 0.8562 | 0.8503 | **0.9943** | 0.8419 |
| JTN-VAE | **1.0** | **0.9996** | 0.9965 | **0.1009** | 0.8551 | 0.8493 | **0.976** | 0.9143 |
| MAM-SMILES | 0.7192 | **0.9999** | 0.9978 | 0.1264 | 0.8557 | 0.8499 | **0.9763** | 0.9485 |
| MAM-SELFIES | **1.0** | **0.9999** | 0.997 | 0.0943 | **0.8684** | **0.8625** | 0.894 | 0.9155 |

**Comparing MAM with SOTA on MOSES molecule generation**

We compare the quality of molecules generated by MAM with standard baselines and state-of-the-art methods in Table 8 and Figure 19. Details of the baseline methods are provided in [46]. MAM-SMILES/SELFIES represents MAM trained on SMILES/SELFIES string representations of molecules. MAM performs either better or comparable to SOTA molecule generative modeling methods. The major advantage of MAM and AO-ARM is that their order-agnostic modeling enables generation in any desired order of the SMILES/SELFIES string (or molecule sub-blocks).

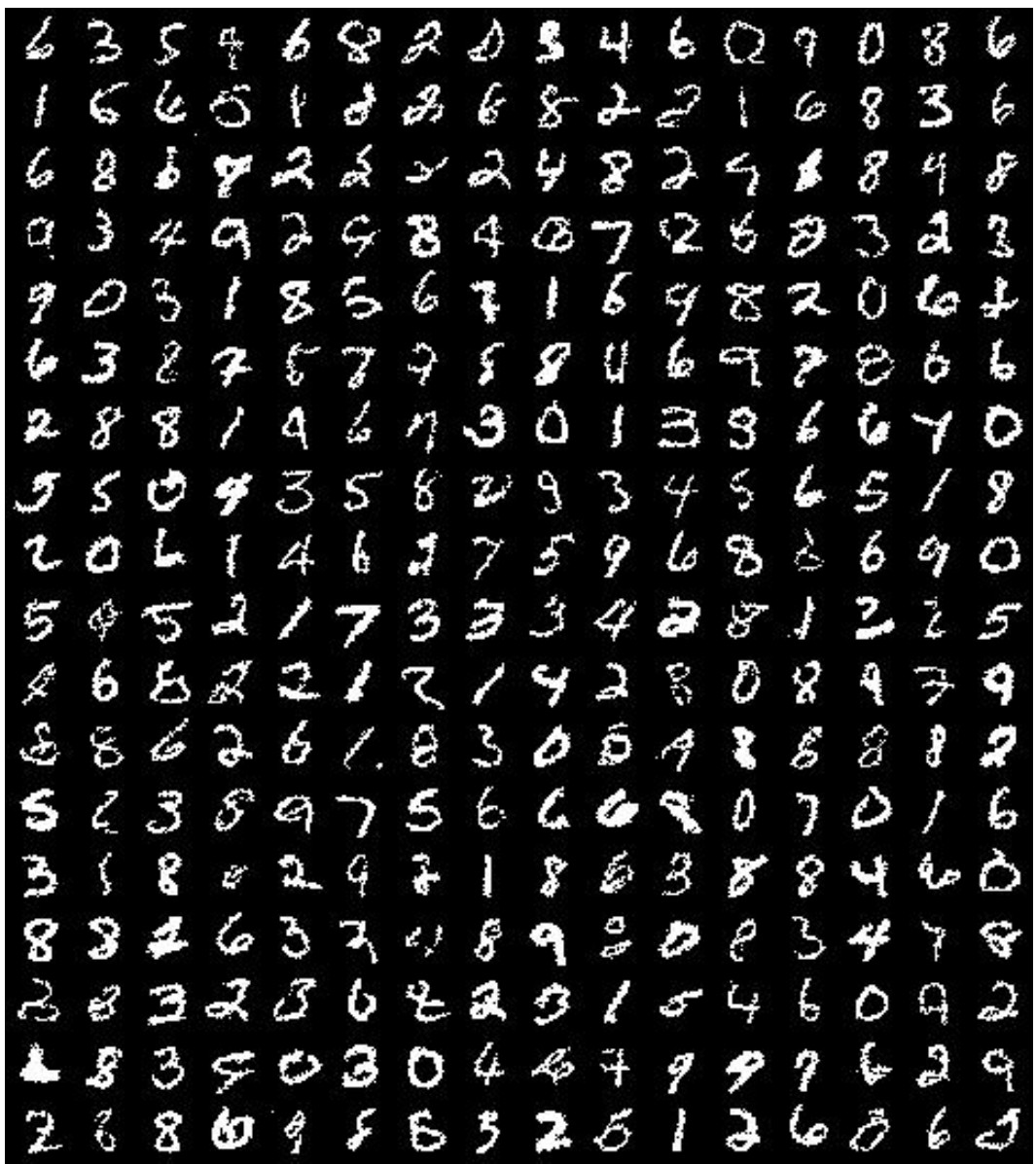

**Figure 18:** Generated samples: Binary MNIST

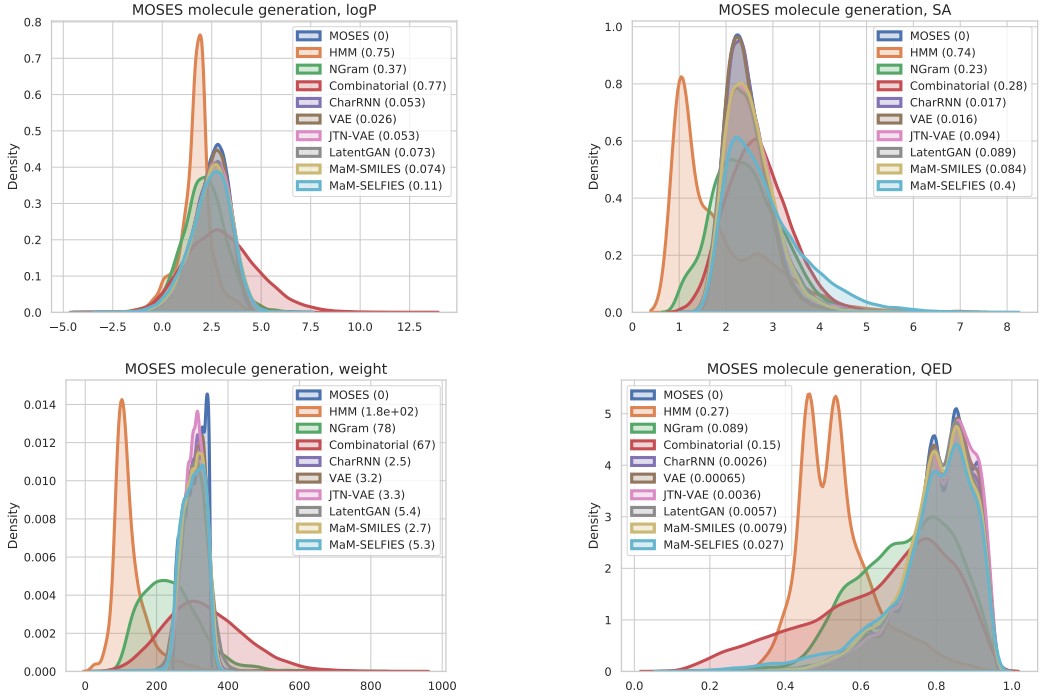

**Figure 19:** KDE plots of lipophilicity (logP), Synthetic Accessibility (SA), Quantitative Estimation of Drug-likeness (QED), and molecular weight for generated molecules. $30,000$ molecules are generated for each method.

**Generated molecular samples**

Figure 20 and 21 plot the generated molecules from MAM-SMILES and MAM-SELFIES.

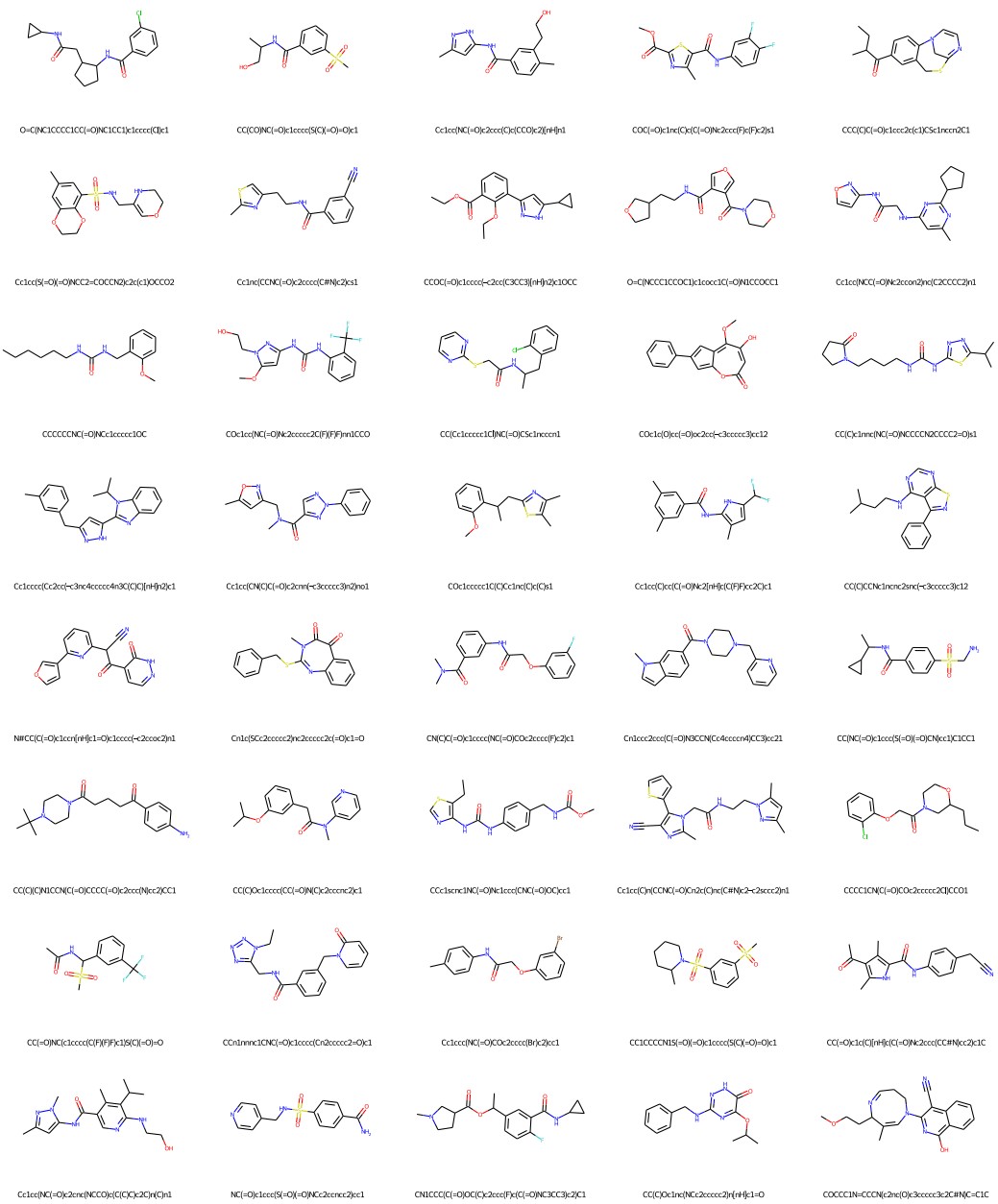

**Figure 20:** Generated samples from MAM-SMILES: MOSES

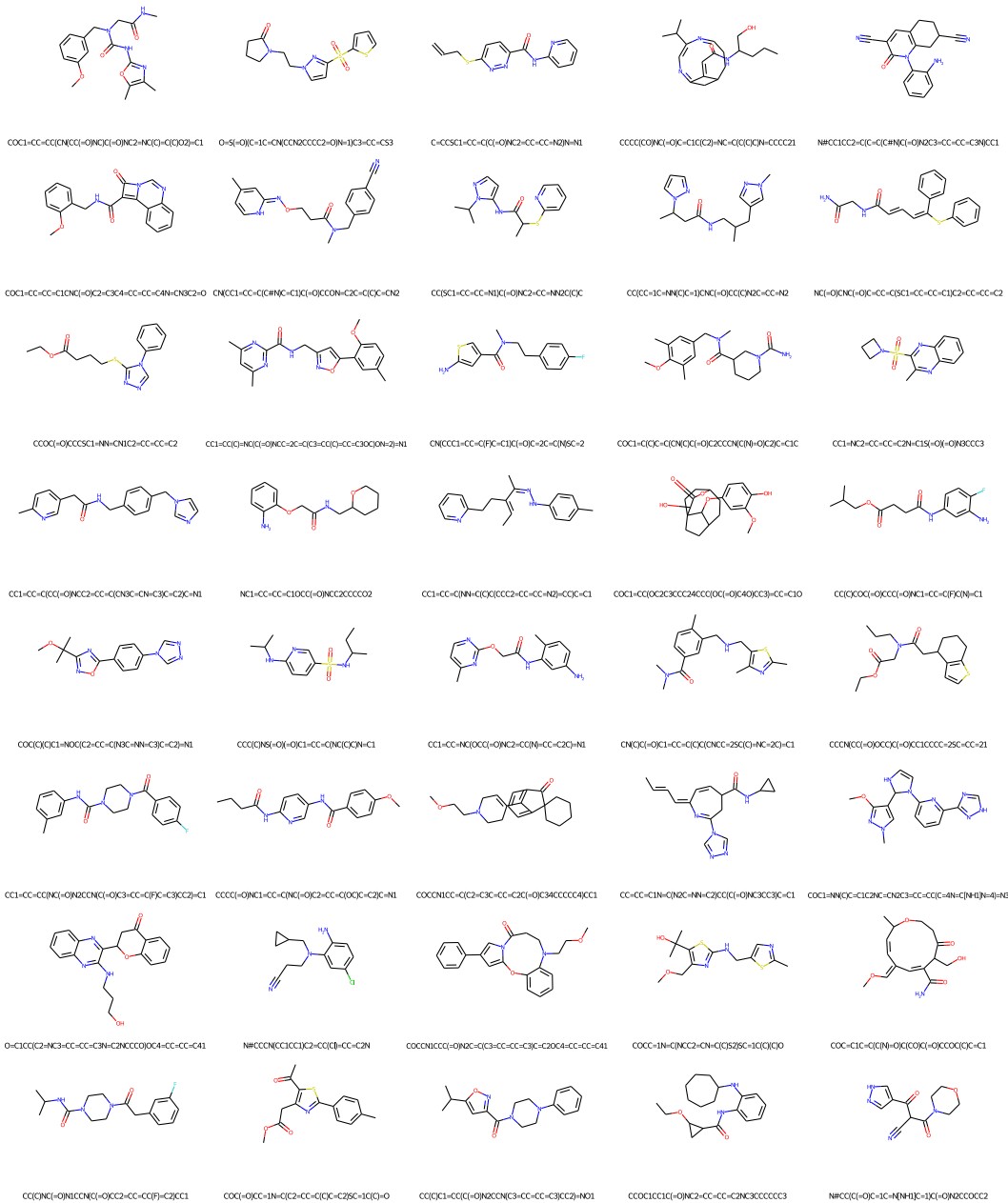

**Figure 21:** Generated samples from MAM-SELFIES: MOSES

### C.6 ADDITIONAL RESULTS ON TEXT8

**Samples used for evaluating likelihood estimate quality**

We show an example of a set of generated samples from masking different portions of the same text, which is then used for evaluating and comparing the likelihood estimate quality. Their log-likelihood calculated using the conditionals with the AO-ARM are in decreasing order. We use MAM marginal network to evaluate the log-likelihood and compare its quality with that of the AO-ARM conditionals.

```
Original text:
the subject of a book by lawrence weschler in one nine nine five
entitled mr wilson s cabinet of wonder and the museum s founder david
wilson received a macarthur foundation genius award in two zero zero
three the museum claims to attract around six
```

```
Text generated from masking out 50 tokens:
 the_su_je_t of a b_ok_by_la_r_nce _es_h___ _n o__nine n_ne five entitled
mr_wilson s_cabinet of wonder and the museum s founder __vid w_l_o_
r__eive_ a macarthur fou__a__on _e___s_awa_d in two _ero z_r_ _hree _he
museum c_aims _o attr_ct ar_u_d s__
the subject of a book by lawrence heschell in one nine nine five
entitled mr wilson s cabinet of wonder and the museum s founder david
wilson received a macarthur foundation dennis award in two zero zero
three the museum claims to attract around sev
```

```
Text generated from masking out 100 tokens:
_the_su_je_t _f __b_k__y_l_r_nc_ _es_h____n o__nine n_ne five_
enti_l_d mr_wil_o_ __c_b__et of wond_r an_ _h_ mu_eu_ s f_u_der___vid
_w_l______eive_ a_maca_thur f_u_a__n _e____a_a_d __ two _er_ z_r_ _h_ee____
museum c_a_ms__o __tr_ct ar_u__ ___
the subject of a book by lawrence bessheim in one nine nine five
entitled mr wilson s cabinet of wonder and the museum s founder david
wilson received a macarthur foundation leaven award in two zero zero
three the museum claims to detract around the
```

```
Text generated from masking out 150 tokens:
_the__u___t_f _________l_r_nc_ _es_h____n o__n__e n_ne__ive_
e_ti_l_m__wil______c___et of won___ an__________ s__u_der___vid_
w________eiv__ a__a_a_th_______a__n _e____a______ two__e__ z_r_ ___e_____ _use_m
c_a_ms___ __tr_ct_a_____ ___
the tudepot of europe de laurence desthefs in one nine nine five
entitled mr wild the cabinet of wonder anne cedallica s founder david
wright received arnasa the culmination team sparked in two zero zero
three the museum claims to retract athlet c a
```

```
Text generated from masking out 200 tokens:
_t__________f _________l__r____ ______ ____o______e_n__iv__
e__i_l__ ___wil______c____t__ w_____a__________________der____d_
w_____ ___e_______a_a_________a__n_e____a______t____________
___e_____ _u_e__c_a__s___ __r_c_a_____ ___
the builder of the pro walter a a e sec press one nine nine five
esciele the wild men convert of wark flax notes the world undergroand
whirl spiken america ascent and martin decree a letter to the antler s
default museum chafes in america ascent vis
```

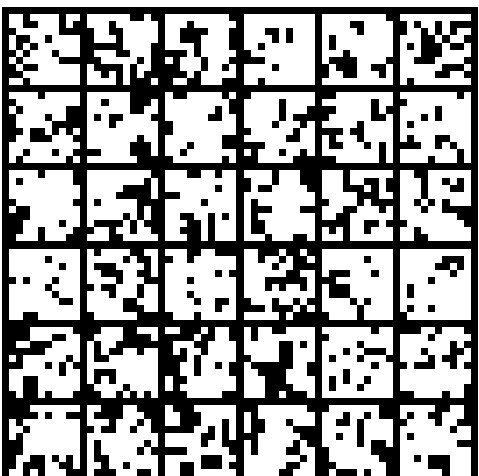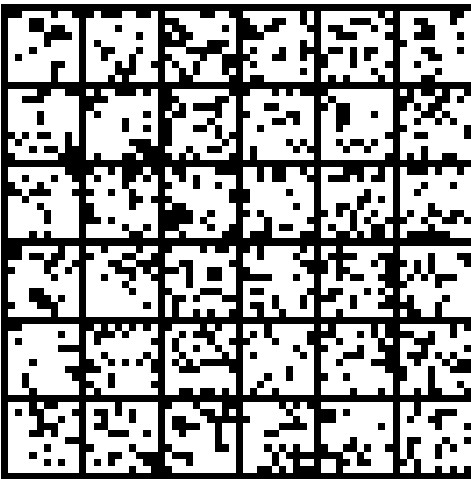

**Figure 22:** Samples: $10 \times 10$ Ising model. Ground truth (left) v.s. MAM (right).

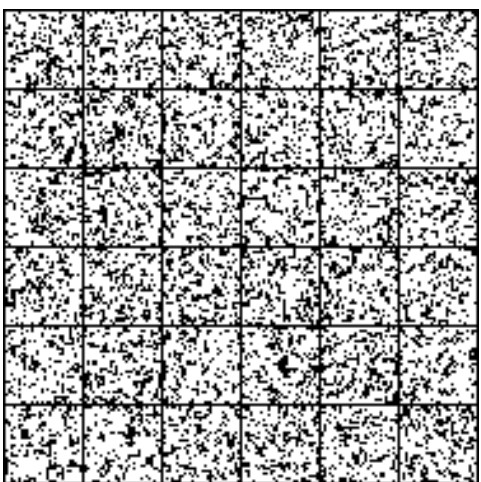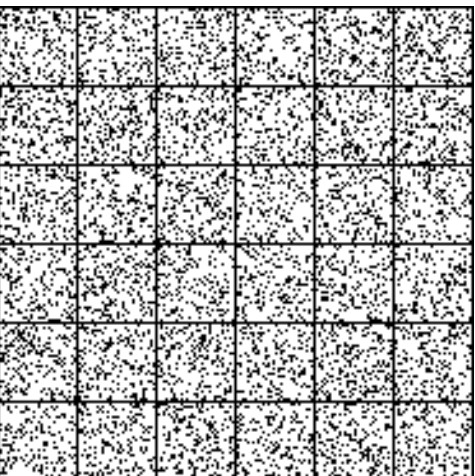

**Figure 23:** Samples: $30 \times 30$ Ising model. Ground truth (left) v.s. MAM (right).

### C.7 ADDITIONAL EXPERIMENTS ON ISING MODEL

**Generated samples**

We compare ground truth samples and MAM samples in Figure 22 and 23.

### C.8 ADDITIONAL EXPERIMENTS ON MOLECULE TARGET GENERATION

**Target property energy-based training on lipophilicity (logP)**

Figure 24 and 25 show the logP of generated samples of length $D = 55$ towards target values $4.0$ and $-4.0$ under distribution temperature $\tau = 1.0$ and $\tau = 0.1$. For $\tau = 1.0$, the peak of the probability density (mass) appears around $2.0$ (or $-2.0$) because there are more valid molecules in total with that logP than molecules with $4.0$ (or $-4.0$), although a single molecule with $4.0$ (or $-4.0$) has a higher probability than $2.0$ (or $-2.0$). When the temperature is set to much lower ($\tau = 0.1$), the peaks concentrate around $4.0$ (or $-4.0$) because the probability of logP value being away from $4.0$ (or $-4.0$) quickly diminishes to zero. We additionally show results on molecules of length $D = 500$. In this case, logP values are shifted towards the target but their peaks are closer to $0$ than when $D = 55$, possibly due to the enlarged molecule space containing more molecules with logP around $0$. Also, this is validated by the result when $\tau = 0.1$ for $D = 500$, the larger design space allows for more molecules with logP values that are close to, but not precisely, the target value.

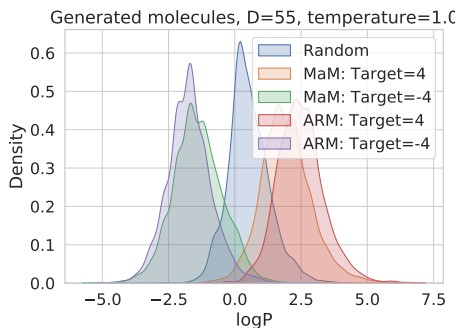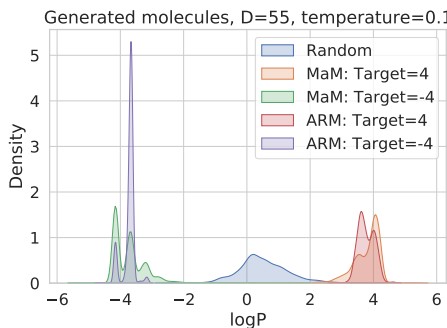

**Figure 24:** Target property matching with different temperatures. 2000 samples are generated for each method.

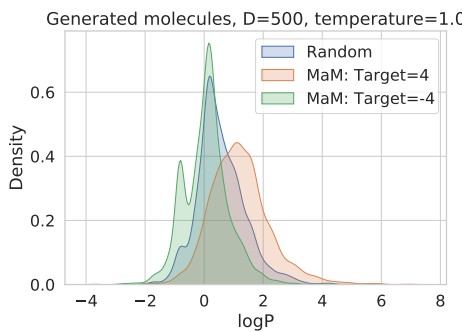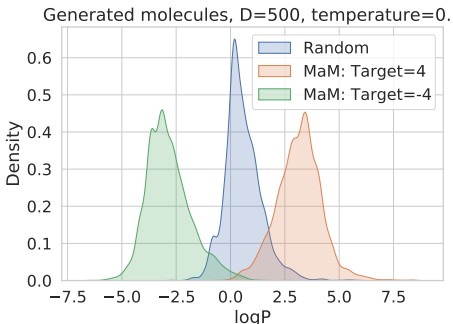

**Figure 25:** Target property matching with different temperatures. 2000 samples are generated for each method.

**Conditionally generated samples**

More samples from conditionally generating towards low lipophilicity (target $= -4.0$, $\tau = 1.0$) from user-defined substructures of Benzene. We are able to generate from any partial substructures with any-order generative modeling of MAM. Figure 26 shows conditional generation from masking out the left 4 SELFIES characters. Figure 27 shows conditional generation from masking the right $4 \sim 20$ characters.

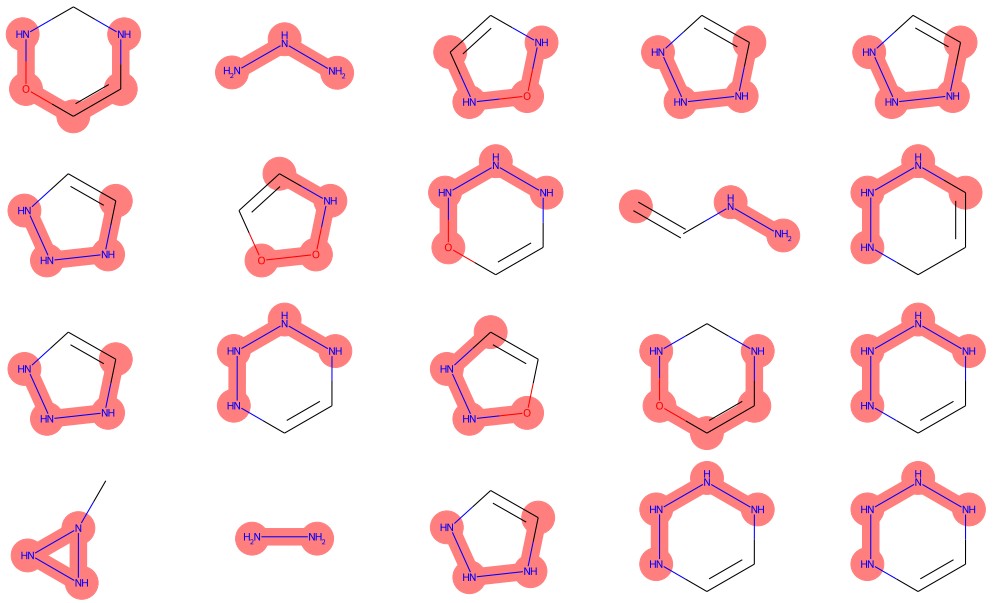

**Figure 26:** Generated samples from masking out the left 4 SELFIES characters of a Benzene.

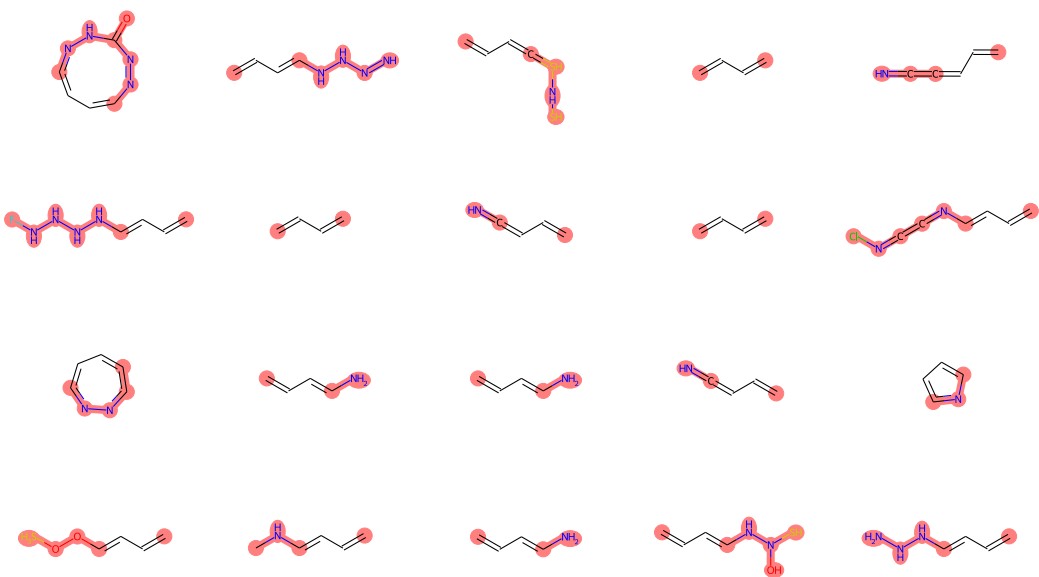

**Figure 27:** Generated samples from masking out the right 4-20 SELFIES characters of a Benzene.

## D  LIMITATIONS AND BROADER IMPACTS

The marginalization self-consistency in MAM is only softly enforced by minimizing the squared error on subsampled marginalization constraints. Therefore the marginal likelihood estimate is not guaranteed to be always perfectly valid. In particular, as a deep learning model, it has the risk of overfitting and low robustness on unseen data distribution. In practice, it means one should not blindly apply it to data that is very different from the training data and expect the marginal likelihood estimate can be trusted.

MAM enables training of a new type of generative model. Access to fast marginal likelihood is helpful for many downstream tasks such as outlier detection, protein/molecule design or screening. By allowing the training of order-agnostic discrete generative models scalable for energy-based training, it enhances the flexibility and controllability of generation towards a target distribution. This also poses the potential risk of deliberate misuse, leading to the generation of content/designs/materials that could cause harm to individuals.

