# OpenReview forum: "Generative Marginalization Models"
_ICLR.cc/2024/Conference — Submitted to ICLR 2024_

### Official Review · Reviewer_kkqm · 2023-10-29

**Soundness:** 3 good
**Presentation:** 3 good
**Contribution:** 3 good
**Rating:** 6
**Confidence:** 4

**Summary:**

The paper presents Marginalization Models (MAMs) as a novel approach to generative modelling for high-dimensional discrete data. MAMs offer scalability, tractable likelihood calculations, and efficient evaluation of marginal probabilities. They support any-order generative models and excel in energy-based training. Experimental results demonstrate MAMs' effectiveness across diverse data types, emphasizing their significant speedup in marginal probability evaluation and their capacity to handle high-dimensional problems, surpassing previous methods.

**Strengths:**

The paper introduces a novel family of generative models that offer both computationally feasible marginalisation and scalability for generating high-dimensional discrete data.

**Weaknesses:**

My primary concern centres around the aspect of marginalization self-consistency. I'm sceptical about whether the soft constraint presented in equation 7 can effectively ensure self-consistent marginalization. The paper lacks reports on real-world applications that involve marginal evaluation. Is it possible to validate the effectiveness of the marginal constraint in some toy examples, like training the model on a predefined  Gaussian distribution with a known marginal density and directly testing it using the mean squared error?

**Questions:**

- To implement the self-consistency constraint outlined in equation 7, you must specify a distribution $q(x)$ for subsampling. How does the choice of the $q$ distribution impact the training process? Is $q(x)$ set to be the data distribution, or can it be any arbitrary distribution?
- When it comes to inference, do you employ equation 6 for sampling, or do you utilize the conditional distribution $p_\phi(x_{\sigma(d)} | x_{\sigma(<d)})$? What’s the difference between these two schemes?
- Point 2) in section 4.3 appears unclear. To train an order-agnostic AutoRegressive Model within the EBM framework, it seems plausible to employ the reinforce gradient estimation method outlined in equation 9. By replacing $\log p_\phi (x)$ with the Monte Carlo estimation of $\mathbb{E}_\sigma \log p_\phi(x|\sigma)$, one could potentially achieve this. Therefore, the statement "ARMs cannot be trained with the expected DKL objective over all orderings simultaneously" is somewhat unclear to me.
- In section 4.3, point 3), I hold a different perspective regarding the efficiency of MAMs compared to ARMs, especially in high-dimensional scenarios. While it's true that in ARM-Full, you do require D feed-forward runs for gradient computation, in MAMs, you also necessitate Gibbs sampling to generate samples from the model. Even if you employ block-wise Gibbs sampling, it still demands multiple steps to guarantee MCMC chain convergence. Hence, I doubt that MAMs also face challenges in high-dimensional problems.
- Auto-regressive diffusion models have demonstrated strong performance on the CIFAR-10 dataset. Is it conceivable that MAM could also yield promising results on CIFAR-10?
- Could you offer additional examples to illustrate why we should be concerned with any-order AutoRegressive models? Do any-order ARMs exhibit superior performance compared to their order-specified counterparts?

---

> ### Author Response · Authors · 2023-11-17
>
> Thank you for your comprehensive review and valuable feedback on our paper. We really appreciate your time and efforts dedicated. We are glad to hear that you found the idea to be novel. We are pleased to address your concerns and questions to better elucidate the contributions and capabilities of MaMs.
>
> - **Why Any-Order auto-regressive models, v.s. fixed order?**
>     - The test NLL achieved by AO-ARM are often just slightly worse than fixed-order ARM, since it is solving a harder problem. But they are interesting and useful due to the flexibility in sampling. For problems without natural ordering, such as molecules, proteins, materials, AO-ARMs allow full control of the generation ordering, and arbitrary conditional generation that fixed-order ARM cannot. See EvoDiff [1] for a recent example in controllable protein design. We have also provided an experiment on molecule string conditional generation (**Figure 7, 24, 25**). Even for tasks such as languages with an ordering, AO-ARMs allow us to do in-fillings and re-edits on existing text, which is useful for iteratively improving and more customizable than current language models.
>     - Access to marginals further increase the flexibility in generation, such as sampling multiple variables at one step (see **general response C1.2 Experiment 2**). Marginals also allow for much better scalability in evaluating marginal likelihood, which is crucial for test time inference of $\log p$ (such as whether a new protein sequence is better in $\log p$ than its counterpart) and scalable training in energy-based setting.
> - **Concerns about marginalization self-consistency**
>
>     Please refer to **general response C1** for details**.** Thanks for your suggestions. We conducted extra experiments to validate self-consistency on a synthetic problem with known ground truth PMF. We also measured self-consistency on real-world problems (see C1.2).
>
> - **Choice of $q$ for sampling self-consistency**
>
>     Please refer to **general response C2.**
>
> - **Sampling with conditionals $p_\phi$ v.s. marginals in eq.(6)**
>     - Please refer to **general response C1.2 Experiment 2.** We conducted extra ablation studies of the two sampling approaches. We have also experimented sampling with different number of variables at one step using marginals.
>     - In **general response C1.1**, we additionally provided a synthetic problem that purely learns marginals and sample using marginals.
> - **Why not train AO-ARM with MC estimation of the gradient in energy-based training?**
>
>     Please refer to **general response C3.**
>
> - **Block-wise Gibbs sampling still demand multiple steps to guarantee MCMC chain convergence.**
>
>     Please refer to **general response C4.**
>
> - **Promising results on CIFAR-10?**
>
>     We followed your suggestion and conducted experiments. Indeed MaM can yield promising results on CIFAR-10. Please refer to **general response C5** for details.
>
>
> We hope this addresses your concerns and answers your questions, and we’re happy to follow up on any questions or concerns in more detail.
>
> [1] Alamdari, Sarah, Nitya Thakkar, Rianne van den Berg, Alex Xijie Lu, Nicolo Fusi, Ava Pardis Amini, and Kevin K. Yang. "Protein generation with evolutionary diffusion: sequence is all you need." *bioRxiv* (2023): 2023-09.

---

> ### Author Response · Authors · 2023-11-21
>
> Dear Reviewer kkqm,
>
> Thanks again for your comprehensive review and valuable feedback. In our previous response, we have conducted additional experiments (on both predefined distribution and real-world distribution) following your suggestions to address your concerns centered around “marginalization self-consistency”. We have also answered the questions raised in your review. In light of our response, we would appreciate if you would consider re-evaluating our submission. Please let us know if there are further questions we can address before the rebuttal period ends.

---

> > ### Comment · Reviewer_kkqm · 2023-11-21
> > **Response to the authors**
> >
> > Thanks for your great effort in responding and revising. Could you show me what the generated image on CIFAR-10 looks like?

---

> > > ### Author Response · Authors · 2023-11-21
> > >
> > > Thanks for acknowledging our response. We have just updated the submission pdf with images generated on CIFAR-10, which include both conditional generation and unconditional generation. Results are consistent with autoregressive models such that AO-ARMs [4] and MAC [3].

---

> > > > ### Comment · Reviewer_kkqm · 2023-11-21
> > > > **Response to the authors**
> > > >
> > > > Thanks for your effort in adding the generated image on CIFAR-10. The results look good. Overall, the paper looks good to me in terms of the idea of ensuring marginalisation constraint in sequential generative models. However, as pointed out by reviewers sWKU and G1Dh, I am still concerned about the effectiveness of such soft constraints. This concern is not raised by the experimental results, but by the less of theoretical guarantee of MaM compared to probabilistic circuits. In this regard, I decided to keep my score unchanged and lean towards accepting the paper.

---

> ### Author Response · Authors · 2023-11-21
>
> Thanks for acknowledging that the idea of ensuring marginalization constraint is interesting and supported by experiments. We appreciate your feedback and being supportive of accepting our paper.
>
> We would also hope to highlight that as a NN-based approximate inference model, our paper's focus is on how modeling marginals unlock new capabilities beyond previous NN-based methods, while maintaining strong generative modeling performance. The new capabilities include:
> - Scalable training under energy-based setting. This was not possible for ARMs due to reasons explained in Section 4.3. The optimizaion of marginalization self-consistency in this setting shares similar spirit with minimizing Bellman residue in actor-critic based methods for reinforcement learning. Although theoretically the equality is not guaranteed, but it enables rewards (energies in this case) to be propagated and make training scalable on high-dimensional problems.
> - Significantly faster marginal inference for comparing $\log p$'s
> - Flexible sampling of multiple variables in one step using marginals
>
> all due to directly modeling the marginals. And those capabilities *do not require marginal estimates to be exactly self-consistent*, as shown in the experiments. As discussed in our response with reviewer sWKU, there is a trade-off between exact marginalization and NN-based soft marginalization. We think it is equally worth exploring in both directions.
> Designing neural networks that inherently satisfy the self-consistency would be a very interesting yet challenging research problem, however it falls beyond the scope of this paper.
>
> Thank you again for your valuable feedbacks and suggestions that help improve our paper.

---

### Official Review · Reviewer_CcDR · 2023-10-30

**Soundness:** 3 good
**Presentation:** 3 good
**Contribution:** 3 good
**Rating:** 6
**Confidence:** 3

**Summary:**

The paper develops marginalization models (MAMs) for modelling all marginals and conditionals for discrete data. It introduces a consistency loss that is combined with either a maximum likelihood objective or an energy-based objective. MAMs require only a single forward pass for computing likelihoods in energy-based learning setups, in contrast to autoregressive models (ARMs). On MLE tasks, the method leads to similar NLL as any-order ARM, but with much smaller marginal inference time. Similarly, for energy-based learning task, the suggested approach performs similar to ARM, but with significant speedup.

**Strengths:**

The paper is generally well written and easy to follow. It addresses an important question.

The idea to introduce a scalable optimisation term that encourages marginalisation self-consistency is new as far as I am aware.

The approach seems to be applicable broadly (I am not sure how well this extends to non-discrete data) and is illustrated on challenging problems. Various experiments on both maximum likelihood training illustrates that it performs comparable with any-order ARMs, while being significantly faster.

**Weaknesses:**

In Section 4.3 2), the authors argue that if the model is not perfectly self-consistent, this poses an issue for ARMs for energy-based setups. As MAMs will likewise not be perfectly self-consistent, why is this not an issue for them? In particular, it is not clear to me that the Gibbs sampling procedure then leads to samples from $p_{\theta}$. Are you adjusting via importance sampling in the experiments?

The paper often assumes that neural networks are universal approximators. It has not become clear why this is a practical assumption for the used architectures as the number of marginal constraints scales badly with K and D. Does this consistency loss actually go to zero in the experiments?

**Questions:**

Is there a reason why you choose the squared difference of the log densities to be matched due to marginalisation constraints over samples from q? If these distributions should coincide, why not use a divergence between them such as their KL?

Can you clarify when the ‘generative’ conditional eq. (6) is used vs. the learnable conditionals with parameter $\phi$?

Can you clarify why a high correlation with AO-ARM-E is a sensible evaluation measure?

---

> ### Author Response · Authors · 2023-11-17
>
> Thank you for your thoughtful review and constructive feedback on our paper. We are glad to hear that you found the paper well-written and the idea of introducing a scalable optimization term for marginalization self-consistency novel. We address below the concerns and questions you have raised.
>
> - **Concerns about training with persistent Gibbs sampling**
>
>     Please refer to **general response C4**.
>
> - **Assumption of neural nets as universal approximators for self-consistency**
>     - Your concern about the practicality of the assumption that neural networks are universal approximators is valid. We make this assumption based on literature and empirical evidence that deep neural networks *can approximate a wide range of functions if given a scalable training objective* (for example the most recent successes include modeling conditional distribution in LLMs, learning score functions in diffusion model). In synthetic example with $D=32$, $K=2$, we observe that self-consistency are almost perfectly enforced with a MLP. Please refer to **general response C1.1.**  In real-world problems, the consistency loss does not reach absolute zero but approaches a small value. Please check **general response C1.2** on how well the self-consistency are enforced and how neural network capacity is required for learning marginals.
>     - To clarify, the assumption is only required for the two-stage efficient training to be theoretically sound. If we drop the assumption, we can still train by using a regularized objective, i.e. $\min_{\theta,\phi} - \mathbb E_{ {x} \sim p_{\text{data}}} \mathbb E_{\sigma \sim \mathcal U\left(S_D\right)} \sum_{d=1}^D \log p_\phi \left( x_{\sigma(d)} \mid x_{ \sigma(<d)} \right) + \lambda \mathbb E_{ x \sim q(x)} \mathbb E_{\sigma \sim \mathcal U\left(S_D\right)} \mathbb E_{d \sim \mathcal U (1, \cdots, D)} \left( \log [p_\theta \left( x_{\sigma(<d)}\right) p_\phi\left( x_{\sigma(d)} \mid x_{\sigma(<d)}\right) ]-\log p_\theta\left( x_{\sigma(\leq d)}\right)\right)^2$This training procedure should achieve the same performance, as long as the first term is approximated with conditionals instead of $\log p_\theta$.
> - **Why** **Choosing squared distance of $\log p$ instead of KL divergence for self-consistency**
>
>     We choose squared distance of $\log p$ so that $q(x)$ can be flexibly specified to reflect the distribution to perform marginal inference on. Please refer to **general response C2** on how $q(x)$ can be set. KL divergence is a good suggestion. The idea of using KL divergence is intriguing. However it is relatively not straightforward to define what is $p$ and $p^\prime$ in $D_\text{KL}(p\parallel p^\prime)$, since both $p$ and $p^\prime$ involve marginals $p_\theta$, the gradient might not be scalable to approximate. If we fix $p^\prime$ to be $p_\theta\left( x_{\sigma(\leq d)}\right)$but detach the gradient and set $p$ to be $p_\theta\left( x_{\sigma(<d)}\right) p_\phi\left( x_{\sigma(d)} \mid x_{\sigma(<d)}\right)$. Then one can show that the REINFORCE gradient of $D_\text{KL}(p\parallel p^\prime)$ falls under a special case of the gradient of our current squared distance loss when $q$ is set to $p_\theta$. (Check [1] for a similar argument on connection of squared loss of $\log p$ and KL divergence.)
>
> - **Using ‘generative’ conditionals (based on marginals) v.s. using learnable conditionals**
>     - Please refer to **general response C1.2 Experiment 2.** We conducted extra ablation studies of the two sampling approaches. We have also experimented sampling with different number of variables at one step using marginals.
>     - In **general response C1.1**, we additionally provided a synthetic problem that purely learns marginals and sample using marginals.
> - **Why measuring** $\log p$ **correlation with that from AO-ARM-E**
>     - *Why compare with AO-ARM-E*: In real-world applications, we don’t have a ground truth PMF to compare to, hence we want to pick the best possible marginal likelihood estimate we can get. AO-ARM-E models the distribution $p_\theta$ the best in terms of attaining the best test NLL. Therefore we regard its marginal estimate to be the best quality to compare with.
>     - *Why measuring correlations*: The correlation is sensible as when we perform marginal inference, it is either using it to compare likelihoods of two samples, i.e. $\log p(x)$ v.s. $\log p(x^\prime)$, or getting normalized conditionals such as $p(x_\mathcal{V} | x_\mathcal{U})  = \frac{p([x_\mathcal{V}, x_\mathcal{U}])}{\sum_{x_{\mathcal{U}}}p([x_\mathcal{V}, x_\mathcal{U}])}$. High correlations means these questions can be answered correctly.
>
> We hope this addresses your concerns and answers your questions, and we’re happy to follow up on any questions or concerns in more detail.
>
> [1] Nikolay Malkin, Salem Lahlou, Tristan Deleu, Xu Ji, Edward Hu, Katie Everett, Dinghuai
> Zhang, and Yoshua Bengio. Gflownets and variational inference. arXiv preprint
> arXiv:2210.00580, 2022.

---

> > ### Author Response · Authors · 2023-11-21
> >
> > Dear Reviewer CcDR,
> >
> > Thanks again for your insightful review and valuable feedback. In our previous response, we have included additional experiments and clarifications to address your major concern on “persistent Gibbs sampling” and "universal approximator assumption". We have also provided answers to your other questions. In light of our response, we would appreciate if you would consider re-evaluating our submission. Please let us know if there are further questions we can address before the rebuttal period ends.

---

> > ### Comment · Reviewer_CcDR · 2023-11-23
> > **Response to rebuttal**
> >
> > Thank you for responses that have largely addressed my questions. I intend to keep my weak accept score.

---

> > > ### Author Response · Authors · 2023-11-23
> > >
> > > We sincerely value the time you spent going through our response. We are happy that our response addressed your questions.  Your positive support of our paper is greatly appreciated.

---

### Official Review · Reviewer_sWKU · 2023-10-30

**Soundness:** 2 fair
**Presentation:** 2 fair
**Contribution:** 1 poor
**Rating:** 5
**Confidence:** 4

**Summary:**

Authors propose Marginalization Models (MaMs), generative models for discrete data allowing (approximate) marginal inference. They key idea is to minimize a penalty term aimed at approximately satisfying the marginalization self-consistency constraint (Eq. 5). The model is evaluated on binary images, text, and molecules.

**Strengths:**

None.

**Weaknesses:**

- The paper presents false claims, and the whole modelling framework is not theoretically solid/supported
- By evaluating what authors call "augumented vector representation" we are not really summing-out the unobserved variables, i.e. we are not performing marginalisation. Rather, we are computing a simple p(x) where x belongs to an augmented state space, and therefore there's no guarantee that this satisfy what authors call "marginalization self-consistency".
- The novelty of the work boils down to the constraint in the loss function (cf. Eq. 9)
- The model is not agnostic to variable orderings and, as such, cannot deal with efficient and *exact* marginalization (see first equation in section 4.2).
- As far as I understood, there's no guarantee that the model satisfies the marginalization self-consistency constraint (Eq. 5), and therefore the model compares to any other dealing with approximate marginal inference, such as [1] [2].
- Sampling is still sequential, as standard autoregressive models. This is weird, as a model allowing for proper marginal inference should not have sequential sampling.
- In general, there should be much more focus and discussion on Probabilistic Circuits, as they satisfy the self-consistency constraint by design, with no need for a penalty term, resulting in exact marginal inference (which is the goal of this paper). Furthermore, PCs are not limited on working with discrete variables only, but can handle heterogeneous data. PCs allow one-shot (conditional) sampling, without relaying on a variable ordering as in MaMs. In short, I believe PCs should be treated as the main competitor of this work, but this is not the case.

[1] Shih, Andy, Dorsa Sadigh, and Stefano Ermon. "Training and Inference on Any-Order Autoregressive Models the Right Way." arXiv preprint arXiv:2205.13554 (2022).

[2] Strauss, Ryan, and Junier B. Oliva. "Arbitrary conditional distributions with energy." Advances in Neural Information Processing Systems 34 (2021): 752-763.

**Questions:**

- INTRO: why marginal evaluations in transformer should be O(D)? I would say that O(D) is the evaluation cost of a fully observed sample,
not for partially observed samples (i.e. marginals). Indeed, transformers, as any standard autoregressive models, cannot perform arbitrary marginalisation.

- INTRO: why EBMs should be limited in fixed-order generative modeling? I do not agree with this claim

- I do not agree with most of what is said in paragraph "Energy-based training", e.g. Why do authors say that in this setting there are no data available?
AFAIK, we do not have access to $f$, rather we model $f$, an unnormalized density/PMF.
When mentioning EBMs I think about [1] and the immense literature deriving from it.

[1] LeCun, Yann, et al. "A tutorial on energy-based learning." Predicting structured data 1.0 (2006).


- What MaMs provide that PCs don't?

---

> ### Author Response · Authors · 2023-11-17
>
> We appreciate your thorough review and insightful comments on our manuscript. Your feedback is invaluable in enhancing the clarity and impact of our work. Below, we address each of your concerns to clarify misunderstandings and reinforce the contributions of our model.
>
> - **Marginalization Self-Consistency**
>     - **Theoretical foundation:**
>
>         We understand your concern regarding the theoretical solidity of MaMs. The foundation of MaMs is built on probabilistic principles, i.e. that the model should adhere to the marginalization self-consistency. In training, we make use of neural nets as powerful function approximator to learn marginals that are self-consistent.
>     - **Investigating self-consistency empirically:**
>
>       We have also investigated in depth on how well the marginalization self-consistency is enforced in MaMs and how the soft-consistencies are very useful in practice. Please refer to **general response C1.1** for details.
>     - **Compare with PCs**:
>
>       Probabilistic circuits are very powerful and promising approaches that exhibit great properties such as fast and exact marginalization through smart design of the model’s structure and operations. We fully acknowledge this and have discussed in details in related work.
>          - **neural approximate inference models v.s. PCs:**
>              - As you have pointed out, our model falls under the category of neural generative models that perform approximate inference, and is more comparable with approaches such as AO-ARM [3], Mast-tuned ARM[1] and Arbitrary conditional energy models (ACE)[2].
>              - Compared to PCs, the neural approximate inference approaches do not have the exact marginalization properties but have better flexibility in modeling complex distributions. This has been shown in the generative models literature and in our paper. We have included more detailed discussion on this (in **Appendix A.4**) about trade-offs between exact marginalization v.s. approximate marginalization.
>         - In experiments, we compare with **both** **neural models** that perform approximate inference and **PCs** that perform exact inference (**Table 1**). PCs have great performance in marginal inference time and is the only one that performs exact marginal inference. In terms of generative modeling performance, there remains a gap between PCs and NN-based methods. On larger problems such as Text8 and molecules, we focus on NN-based approaches, since we do not find PC models designed for text or molecules.
>             - We have also included MAC [1] following your suggestions. (See **general response C5** for details.) [2] focus on continuous data and mixed data, hence we do not directly compare. We also compared with SOTA discrete generative model such as AO-ARM [1] and GFlowNets.
>         - Last, we want to clarify that the perspective of this paper is to propose a neural-network based generative model based on learning marginals. MaMs unlock new capabilities beyond previous NN-based methods, while maintaining strong generative modeling performance. The new capabilities include: 1) significantly faster marginal $\log p$ inference, 2) scalable training under energy-based setting, and 3) flexible sampling of multiple variables in one step using marginals, all due to directly modeling the marginals.

---

> > ### Author Response · Authors · 2023-11-17
> >
> > - **Sampling is still sequential**
> >     - MaMs additionally allow sampling multiple variables of different sizes in arbitrary order at each step, which is more flexible than ARM. Please refer to **general response C1.2**. We acknowledge that PCs allow one-shot conditional sampling due to the nice structures of the sum and product nodes. We have included this discussion in the updated manuscript.
> >
> > - **Energy-based setting**
> >     - **“Why AO-ARMs are limited in fixed-order modeling”:**
> >
> >         Please refer to **general response C3**.
> >
> >     - **“EB setting definition not clear”:**
> >
> >         Thank you for the suggestion, the energy function indeed can be inferred from data (in the case of energy-based models) or pre-defined (in cases that we know the ground truth, such as materials, proteins). We want to use EB training to describe the second case. Since in the first case of EBM, training an EBM is still using MLE objective, i.e. $\mathbb E_{x \sim p_{\text {data }}(x)}\left[\log p_{\theta}( x)\right]$. Sorry for the confusion, we have made clarifications in the updated manuscript.
> >
> >
> > We hope this addresses your concerns and answers your questions, and we’re happy to follow up on any questions or concerns in more detail.
> >
> > [1] Shih, Andy, Dorsa Sadigh, and Stefano Ermon. "Training and Inference on Any-Order Autoregressive Models the Right Way." arXiv preprint arXiv:2205.13554 (2022).
> >
> > [2] Strauss, Ryan, and Junier B. Oliva. "Arbitrary conditional distributions with energy." Advances in Neural Information Processing Systems 34 (2021): 752-763.
> >
> > [3] Emiel Hoogeboom, Alexey A. Gritsenko, Jasmijn Bastings, Ben Poole, Rianne van den Berg,
> > and Tim Salimans. Autoregressive diffusion models. In 10th International Conference on
> > Learning Representations, 2022

---

> > > ### Author Response · Authors · 2023-11-19
> > >
> > > We would like to follow up on the concern of **comparing with probabilistic circuits**.
> > >
> > > As mentioned in our response, we have included more discussion on the comparison between neural network based generative models v.s. probabilistic circuits. Our paper included PCs as a baseline in the MNIST experiment. For Text8 and Molecule Sets, we are not able to find implementation of PCs that are well-suited for the problem. As far as we understand, PCs have not been applied to energy-based training either. If you think our comparison and discussion on PCs are inadequate, please let us know. We are happy to incorporate your suggestions and address any concerns you might have.
> > > Thanks again for your valuable feedback.

---

> ### Comment · Reviewer_sWKU · 2023-11-19
>
> Thanks for your detailed rebuttal. Given the clarifications in the current state of the paper I'm increasing my score from 3 to 5 (although I feel a 4 to better reflect my position). Given the approximate nature of the method proposed and on its limitation to work with discrete data, part of my concerns remain.
>
> I would just like some more clarifications about the number of discrete state $K$. Can you elaborate a bit more the paragraph "Scalable learning of marginals with conditionals"? Could your proposed solution be enough to test MaMs on the (non-binary) MNIST dataset? If so, could you report results in bpd? The literature is filled of results for MNIST, and as such would help contextualize better the capabilities of your model.
>
> Finally, note that computing marginals could be very important for neural compression [A]. If MaM allows accurate approximation of marginals, why not testing how well it would perform in neural compression tasks?
>
> [A] Anji Liu, Stephan Mandt, and Guy Van den Broeck. Lossless compression with probabilistic circuits. In International Conference on Learning Representations, 2022

---

> ### Author Response · Authors · 2023-11-20
>
> Thanks for your reply and valuable suggestions. We appreciate your acknowledgement of our clarifications. We are happy to clarify on the questions you raised.
>
> - **Number of discrete state $K$**
>
>     We conducted an extra experiment on MNIST ($K=256$) with the approach proposed in Section 4.1, Claim 1 (with the scalable learning objective in (7) proposed when $K$ is large). MaM achieve a test NLL of $0.577$ bpd (using the conditionals), which is better than most SOTA such as Locally Masked PixelCNN ($0.65$). The marginal $\log p$ from MaM have a spearman correlation $0.940$ and a pearson correlation $0.941$ (as compared to $\log p$ from conditionals). We have also conducted an experiment on CIFAR-10 ($D=3\times 32 \times 32$, $K=256$) in this case. MaMs achieve competitive test NLL and (fast) marginal estimates as well, please refer to **general response C4**.
>
> - **Neural compression**
>     - Thanks for the great suggestion and the great reference. Neural compression is indeed an important use-case of exact likelihood models such as PCs, ARMs and Normalization Flows. We tested neural compression with MaM on MNIST-Binary, and the **actual bpd** is $0.154$ which is close to the (theoretical) **test NLL** $0.146$. We will include this result in our updated manuscript.
>     - Since compression algorithms such as rANS require a sequence of PMF and CDF over the variables, MaM will incur similar amount of compute time as autoregressive models, i.e. compute PMF and CDF along every dimension. Hence we did not focus on this task as it is not when MaM should be used instead of models such as ARMs and Flow models. However, by utilizing ideas in MADE [1], it might be possible to use just one NN forward pass to get all the PMF and CDF needed for neural compression. But again there is a trade-off in terms of model expressiveness v.s. efficiency. This is also related to how PCs with structures [A] enable $\mathcal O(\log D)$ evaluations.
> - **Approximate v.s. exact**
>
>     We appreciate the feedback that the method is limited by its approximate nature. We would like to point out that there is a constant trade-off of approximate and exact. By utilizing the approximation power of NNs, we have to give up some nice theoretical properties. In this case, the marginals are soft-consistent instead of being exactly consistent. We also empirically measured how well the self-consistencies are preserved and how the marginals learned can be useful in practice. MaMs unlock **three new capabilities** — 1) significantly faster marginal $\log p$ inference, 2) scalable training under energy-based setting, and 3) flexible sampling of multiple variables in one step using marginals, all due to directly modeling the marginals —  **beyond previous NN-based methods**, while maintaining strong generative modeling performance. We think our work proposed a new idea and provided evidence that learning approximate marginals are worth exploring, in parallel to how PCs with tractate likelihood properties are worth exploring as well. At the moment, they might be best suited for different tasks where marginal inference are required. But there is more hope that by further exploring those ideas, powerful generative models with fast exact likelihood can become a reality.
>
> We are happy to follow up on any further questions or concerns you might have.
>
> [1] Papamakarios, George, Theo Pavlakou, and Iain Murray. "Masked autoregressive flow for density estimation." *Advances in neural information processing systems* 30 (2017).

---

### Official Review · Reviewer_G1Dh · 2023-11-04

**Soundness:** 2 fair
**Presentation:** 3 good
**Contribution:** 2 fair
**Rating:** 5
**Confidence:** 3

**Summary:**

The paper introduces Generative Marginalization Models (MAMs), which are a new type of generative model for high-dimensional discrete data. MAMs address the limitations of existing methods by explicitly modeling all derived marginal distributions, allowing for scalable and flexible generative modeling. MAMs can rapidly evaluate any marginal probability with a single forward pass of a neural network, without requiring accurate marginal inference. MAMs also support scalable training of generative models with arbitrary orderings of high-dimensional problems in an energy-based training setting. The effectiveness of MAMs is demonstrated on various discrete data distributions, including binary images, language, physical systems, and molecules. MAMs significantly speed up the evaluation of marginal probabilities, enabling the modeling of arbitrary orderings of high-dimensional problems that were not achievable with conventional methods in energy-based training tasks.

**Strengths:**

- The authors introduce a new family of generative models called Marginalization Models (MAMs) that have tractable likelihoods and offer scalable and flexible generative modeling.
  - MAMs allow for fast evaluation of arbitrary marginal probabilities with a single forward pass of the neural network, overcoming a limitation of autoregressive models.
  - The proposed model supports scalable training of any-order generative models for high-dimensional problems under the energy-based training setting.
  - MAMs achieve significant speedup in evaluating marginal probabilities.
- The effectiveness of the proposed model is demonstrated on various discrete data distributions, including binary images, language, physical systems, and molecules.
- The authors identify an interesting connection between generative marginalization models and GFlowNets, showing their equivalence under certain conditions in A.3 in the appendix.

**Weaknesses:**

- When I read the introduction, I expected to find a way to rigorously guarantee self-consistency constraints, but in fact I was somewhat underwhelmed because the actual way is only to impose soft constraints during optimization.
- The experimental results are not so good. In many experiments, the evaluation with NLL is slightly worse than the baseline, which does not fully demonstrate the superiority of the MAM.
- The effectiveness of two-stage training has not been adequately explained. At least, there should be an experimental comparison with the case without the two-stage training.

**Questions:**

- I do not understand why two-stage training is effective. Do the authors have any hypothesis on it?
- What is the definition of the marginal inference time in Tables 2, 3, and 4?

---

> ### Author Response · Authors · 2023-11-17
>
> Thank you for the time and efforts devoted to reviewing our paper. We appreciate the valuable feedbacks and hope to clarify on the questions and concerns. We have conducted extra experiments to support our clarifications. We have also incorporated the valuable feedbacks into the paper.
>
> - “**Self-consistency is only enforced softly”**
>     - Please refer to **general response C1.**
> - **“Definition of marginal inference time”**
>
>     We are happy to make clarifications. Marginal inference here means given a $x_\mathcal{S}$ (for example $x_\mathcal{S}= (0,1,1,1,1,1,?,?,?)$) evaluate the marginal probability under the learned generative model, i.e. $p_\theta(x_\mathcal{S})$. With MaM, we use the marginal network $\theta$ to evaluate $p_\theta(x_\mathcal{S})$ with a single NN forward pass. As with AO-ARM, it requires evaluating a sequence of conditionals (NN forward passes) to get the marginal of $p_\theta(x_\mathcal{S})$. Those marginals are compared to the marginal from the best likelihood estimation model (AO-ARM-E) by measuring Spearman’s and pearson correlations.
>
> - **“Experiment results are not good”**
>
>     We believe this is a misunderstanding on how methods should be compared to each other. The main merit of MaMs is in unlocking new capabilities beyond previous methods, while maintaining strong generative modeling performance. The new capabilities include: 1) significantly faster marginal $\log p$ inference, 2) scalable training under energy-based setting, and 3) flexible sampling of multiple variables in one step using marginals, all due to directly modeling the marginals.
>
>     - **Comparable generative modeling performance**: In MLE setting, the fair comparison of test NLL will be MaM with AO-ARM-Single, since that is how data is generated (using a single random ordering). Since both methods have learnable conditionals, they achieve the same NLL. The AO-ARM-Ensemble model uses an ensemble of orderings for evaluating a better quality marginal likelihood, which is used only as a reference. Because in reality, AO-ARM-Ensemble cannot be used for generation. We are sorry for the confusion.
>     - **Significantly faster marginal** $\log p$ **inference** (see last column of Table 1,2,3,4) as compared with all other methods while maintaining its generative modeling performance. Probabilistic circuits (PCs) are also fast in marignal evaluation but their generative modeling capability is not as good as NN-based approaches.
>     - **Training with scalability on high-dimensional problems under EB setting:** For example, Ising-model with $900$ dimensions and molecule strings with $500$ dimensions that other approaches fail to scale to.
>     - **Flexible sampling with marginals**: In **general response C1.2**, we provided additional experiments illustrating the power of marginal estimations in sampling with various block variable sizes that other SOTA methods such as AO-ARM cannot perform.
>
> - **“Two-stage training explanation”**
>
>     Thanks for the suggestion. We are happy to include more explanation.
>
>     - The basic intuition is that the optimal conditionals can be obtained by doing Stage 1 just by maximizing the likelihood. And this is shown by proof of Proposition 1 in Appendix A.1. In practice, we find it most important to use the expectation over conditionals $\mathbb E_{ x \sim p_{\text {data}}} \mathbb E_{\sigma \sim \mathcal U \left(S_D\right)} \sum_{d=1}^D \log p_\phi\left(x_{\sigma(d)} \mid x_{\sigma(<d)}\right)$ for the $\mathbb E_{ x \sim p_{\text {data}}} \log p_\theta( x )$ part of the loss in eq.(8). Otherwise if marginal network is used for this part, we empirically find maximizing likelihood pushes $\log p_\theta( x)$ to high values that break the self-consistency. We propose two-stage optimization since it is theoretically principled and this is more GPU memory efficient (than optimizing $\phi,\theta$ jointly). If there is enough compute, it is possible to train Stage 1 objective and Stage 2 objective jointly, i.e. $\min_{ \theta,\phi} - \mathbb E_{ x \sim p_{\text {data }}} \mathbb E_{\sigma \sim \mathcal U \left(S_D\right)} \sum_{d=1}^D \log p_\phi\left(x_{ \sigma (d)} \mid x_{\sigma(<d)}\right) + \lambda \mathbb E_{x \sim q(x)} \mathbb E_{\sigma \sim \mathcal U\left(S_D\right)} \mathbb E_{d \sim \mathcal U(1, \cdots, D)}\left(\log \left[p_\theta\left(x_{\sigma(<d)}\right) p_\phi\left(x_{\sigma(d)} \mid x_{\sigma(<d)}\right)\right]-\log p_\theta\left(x_{\sigma(\leq d)}\right)\right)^2$
>     - In **general response C1.1**, we also tested just training MaM marginals (no conditionals are learned) on a synthetic problem. In that simple case, training with a regularized objective is totally fine and results in self-consistent marginals.
>
> We hope this addresses your concerns and answers your questions, and we’re happy to follow up on any questions or concerns in more detail.

---

> > ### Author Response · Authors · 2023-11-20
> >
> > We would hope to follow up on if you have any further questions or concerns. In our previous response, we have provided clarifications and conducted additional experiments to address your concerns about **enforcing self-consistency** and **MaM’s performance**. We have also provided clarifications on **two-stage training** and **how MaMs should be evaluated against other methods**.
> >
> > Thank you once again for your valuable feedback We have made significant efforts to incorporate your suggestions and address your concerns with additional experiments. In light of our response, we would appreciate if you would consider re-evaluating our submission and raising your score accordingly. Please let us know if there are further questions we can address.

---

> > > ### Author Response · Authors · 2023-11-22
> > >
> > > Dear reviewer G1Dh,
> > >
> > > We would like to follow up again to see if you have any further concerns. Your concerns on *enforcing self-consistency*, *overall performance* and *two-stage training* are addressed in our response with experimental evidence. In particular, we believe there was some misunderstanding on how MaMs should be evaluated against other methods. We would greatly appreciate if you could check our response and consider re-evaluating our response in light of the responses. Thanks again for your suggestions and efforts in reviewing our paper.

---

### Official Review · Reviewer_2RBk · 2023-11-05

**Soundness:** 3 good
**Presentation:** 4 excellent
**Contribution:** 3 good
**Rating:** 8
**Confidence:** 3

**Summary:**

This paper introduces Marginalization Models (MAMs), a novel class of generative models that brings scalability and flexibility to generative modeling while allowing tractable estimation of the marginal likelihood for any subset of multivariate random variables. MAMs can accommodate both maximum likelihood training and energy-based training, making them exceptionally versatile, particularly when dealing with discrete random variables, such as molecules, or random variables defined by an energy function. Extensive experimental results show the superior efficiency of MAMs compared to autoregressive models.

**Strengths:**

Originality & Significance: The paper proposes a new type of generative models MAMs. It can handle both maximum likelihood training and energy-based training, which is quite promising. The ability to handle such diverse training methods strengthens the significance of this work.

Quality: The work is well-motivated and logically compact. The claim in Sec. 4.1 and discussion in Sec. 4.3 are sound.

Clarity: The paper is generally well-written and easy to follow. Figures are very illustrative.

**Weaknesses:**

One issue with this paper is that it mainly uses small, simple datasets for its experiments. While the results look good with these small datasets, I'm not sure how well the model would work with larger, more complex real-world data. Real-world data is often much bigger and more complicated, presenting a lot of variables and intricacies. For instance, the number of possible orderings scales fractionally with data dimension To really understand if this approach is useful in practice, it would be helpful to test it on bigger and more diverse datasets.

Minor:
- typo in page 4: where $K$ is the number of discrete values $x_d$ can take -> ... discrete values $x_{\sigma(d)}$ can take

**Questions:**

- >In this paper, we focus on the sampling procedure that generates one variable at a time, but marginalization models can also facilitate sampling multiple variables at a time in a similar fashion.

    How can MAMs generate multiple variables simultaneously? It appears that MAMs are limited to generating variables sequentially, as indicated in Equation (5) and (6).

- How does the order in which variables are sampled impact the quality of the samples?

- How does different $q$ (e.g., uniform v.s. $p_{\\mathrm{data}}$) affect the learning of MAMs?

- I'm a bit confused about the sample generation process with MAMs. In Section 3, it mentions using the normalized conditional (Equation (6)) for sampling, but in Section 6.1, it's said that the conditional model $p_{\\phi}$ is used to generate data. Can you clarify this process?

- Given the two-stage training approach for MLE, it seems that the learned conditional model $p_{\phi}$ is independent of the marginalization model $p_{\\theta}$. As mentioned on page 5, $p_{\\theta}$ is described as distilling marginals from conditionals. How does the conditional model, as an AO-ARM model, compare to the AO-ARM baselines?

- In the experiments, how are the NLL (bpd) values estimated for MAM? Are these values the outputs of $p_{\theta}$ or the logarithmic products of $p_{\\theta}$?

---

> ### Author Response · Authors · 2023-11-17
>
> Thank you for the constructive feedback and questions. We appreciate your time in reading our paper in details and catching typos. We have conducted extra experiments to support our clarifications. We have also incorporated the valuable feedbacks into the paper.
>
> - **“Small, simple datasets for evaluation”**
>     - Datasets like MNIST Binary are relatively small. We study on them first since this is a new method and we want to understand it better. Text8 and Molecular Sets (MOSES) are relatively big and widely used for testing generative models, see [1][2] for example. Text8 has 100M characters and sequence is of length $D=250$. MOSES contain ~2M molecular structures, and the sequence length is $D=55$ (for SMILES) or $D=57$ (for SELFIES). We have also included an experiment CIFAR-10 **in general response C5**, which has much higher dimensions ($D=3 \times 32 \times 32, K=256$) than MNIST Binary ($D=28 \times 28, K=2$). MaMs are able to learn marginals effectively and use them for marginal inference.
> - **Sampling and evaluation with MaMs**
>
>     We apologize for not including all the details in our paper.
>
>     - “**How can MaMs generate multiple variables simultaneously?**”: For generating multiple variables simultaneously, in **general response C1.2** we provide more details and conducted experiments to show this. The idea is in similar spirit to eq.(6): evaluate marginals on all possible outcomes over combinations of multiple variables and get the normalized sampling probability, i.e. $p_\theta( x_{s_i} | x_{s(<i)}) = \frac{p_\theta([ x_{s_i}, x_{s(<i)}])}{\sum\nolimits_{ x_{s_i}} p_\theta([ x_{s_i}, x_{s(<i)}])}$, where $s_i$ is the set of multiple variables to sample at this step.
>     - “**How is sampling/generation and evaluation done with MaM?**”:
>         - We can use either the conditional or the marginal for generation and evaluation. The sampling and evaluation of test NLL in the original submission is using conditionals. We have additionally included sampling and evaluation using marginals with different number of variables at one step.  Please refer to **general response C1.2** for details.
>         - The test NLL is evaluated with products of conditionals $p_\phi$ or $p_\theta$ in eq.(6),. The numbers from the output $p_\theta(x)$ on a complete $x$ are not used since they are not exactly normalized probabilities. In experiments (see **general response C1.2 Experiment 1**), we evaluate the quality of $p_\theta$, and show they are highly correlated with the actual $\log p$ from conditionals.
>         - In **general response C1.1**, we additionally included a synthetic task that purely learns marginals and samples using marginals.
>
> - **“How does the order in which variables are sampled impact the quality of the samples?”**
>     - We do not observe much difference, since MaMs are trained by sampling with arbitrary orderings. Same applies to AO-ARMs. For problems with natural ordering, such as language or maybe images, it is observed that training on a fixed ordering gives better results than training on any-orderings, see [1]. But that approach loses the flexibility of generating in any-order.
> - **“How does different q (e.g., uniform v.s. p_data) affect the learning of MAMs?”**
>     - Please refer to **general response C2.**
> - **How does the conditional model, as an AO-ARM model, compare to the AO-ARM baselines?**
>     - They perform relatively the same since both are trained with the same objective $\mathbb E_{ x \sim p_{\text {data }}} \mathbb E_{\sigma \sim \mathcal U \left(S_D\right)} \sum_{d=1}^D \log p_\phi\left(x_{\sigma(d)} \mid {x}_{\sigma(<d)}\right)$, except MaM additionally learn marginals.
>
> We hope this addresses your concerns and answers your questions, and we’re happy to follow up on any questions or concerns in more detail.
>
> [1] Emiel Hoogeboom, Alexey A. Gritsenko, Jasmijn Bastings, Ben Poole, Rianne van den Berg,
> and Tim Salimans. Autoregressive diffusion models. In 10th International Conference on
> Learning Representations, 2022
>
> [2] Shih, Andy, Dorsa Sadigh, and Stefano Ermon. "Training and Inference on Any-Order Autoregressive Models the Right Way." arXiv preprint arXiv:2205.13554 (2022).

---

> > ### Author Response · Authors · 2023-11-21
> >
> > Dear Reviewer 2RBk,
> >
> > Thank you again for the constructive feedbacks. In our previous response, we have conducted additional experiments and made clarifications to address your concern on “using simple datasets”. We've also taken the opportunity to respond to your questions in detail supported by experiments. Please let us know if there are further questions we can address before the rebuttal period ends.

---

### Author Response · Authors · 2023-11-17
**General response with additional ablation studies/experiments**

### **C1: Marginalization self-consistency is not strictly enforced. How effective is it when softly constrained?  (Reviewers 2RBk, G1Dh, CcDR, kkqm)**

- **C1.1 Self-Consistency Check on Checkerboard PMF:**
    - We’ve added new results (in **Appendix B.1**) on a popular synthetic problem *checkerboard* with a well-defined probability mass function. MaM marginalization consistency is measured against the ground-truth and its self-consistency is also measured. On both MLE and EB settings (**Figure 8, 9, 10, 11 in Appendix B.1**), the marginals are approximately consistent (almost perfectly consistent under EB training since there is a target $p$). When regularization is not strong enough ($\lambda=1e2$ with MLE), the (log) marginals are soft-consistent, but they are only shifted by a constant from the ground truth. After normalization, they match very well with ground-truth (**Figure 12**). When used for sampling, both strong-consistent and soft-consistent marginals lead to same performance (**Figure 10**).


        | Training method | PMF-consistency (dark pixels)  | PMF-consistency (light pixels) |
        | --- | --- | --- |
        | Energy-based, $\lambda=1.0$ | $p: 7.67e − 20$, $\log p: 0.0033$  | $p: 3.73e − 30$, $\log p: 0.0076$ |
        | MLE, $\lambda=1.0e4$ | $p:2.5e − 19$, $\log p: 0.533$ | $p: 3e − 28$,  $\log p: 2.5$ |
        | MLE, $\lambda=1.0e2$ | $p:1.82e-07$, $\log p: 25.83$ | $p: 4.87e-23$, $\log p: 188.45$ |
    - We would like to point out that, as long as the learned marginals are consistent with their counterparts (i.e. $p(0, 0, ?, ?)$ v.s. $p(0,1,?,?)$), shifting by a constant does not affect their utility in sampling data or comparing $p(x_\mathcal{S})$’s and computing $p(x_\mathcal{U} | x_\mathcal{V})$. In other words, for most practical utility, NN just need to infer correctly how likelihoods such as  $p(0, 0, ?, ?)$ v.s. $p(0,1,?,?)$ compare with each other. And our proposed scalable training objective based on squared loss of $\log p$’s achieves this quite well for all marginals empirically. Next we examine this claim on real-world cases.
- **C1.2 Self-Consistency Check on MNIST Binary:**

    Without a well-defined ground truth PMF, it is hard to check self-consistency exactly on real-world datasets. We designed two experiments to do this.
    - **Experiment 1: MaM marginals are extremely highly correlated** with the self-consistent marginals computed with the best-performing exact likelihood model (AO-ARM in this case), but much faster to evaluate ($O(1)$ v.s. up to $O(D)$). We check this on both $x$ and subset of $x$ on different applications (**Tables 1,  2, 3, 6**).
    - **Experiment 2:** **Using marginals for simultaneously sampling blocks of variables:** We conducted extra experiments that explore using the learned marginals $\theta$ for sampling with arbitrary number of variables and ordering: $p_\theta ( x_{s_i} | x_{ s(<i) } )  = \frac{ p_\theta ( [ x_{s_i}, x_{s(<i) } ] )} {\sum \nolimits_{ x_{s_i} } p_\theta ( [ x_{s_i}, x_{s (<i) } ] )},$ where $s_i$ is the subset of multiple variables to sample at each step $i$. We tested sampling with $1,2,4,8$ pixels at each step on MNIST Binary. The samples generated are of similar quality (see **Figure 13 in Appendix B.2**). Different sampling procedures achieve similar likelihood on test data too. It is worth noting that, sampling with large block size enables us to trade compute memory for less time spent (due to fewer steps) in model sampling inference, which we find it to be an interesting property of MaM. This illustrates the potential utility of MaMs in efficient flexible generation additional to conditional-based autoregressive models.
    - We also want to touch on **why the NN universal approximation assumption are relevant.** In our MNIST experiment, we find that finetuning on the last layer of the trained conditionals does not give approximately self-consistent marginals (i.e. the self-consistency error remains high). By increasing the capacity of the marginal network (i.e. include more learnable layers), the marginal self-consistency starts to drop and the learned marginals become approximately consistent (MSE less than 0.01 on $\log p$ for eq (7)). “The neural network really wants to learn”[2], but it needs to have the capacity of doing functional approximation and a training objective that is scalable to be trained on.

---

> ### Author Response · Authors · 2023-11-17
> **General response with additional ablation studies/experiments (2)**
>
> ### **C2: Choice of $q$ for sampling marginal consistency objective (Reviewers 2RBk, CcDR, kkqm)**
>
> Thanks for the great clarification question.
>
> - In simple examples (such as *checkerboard)*, it does not really matter, we have tried $p_\text{data}$, $p_\theta$, uniform random, or a mixture of them. All work fairy well given that the problem is relatively easy.
> - In real-world problems, it boils down to what the marginal will be used for at test time. Uniform distribution over $x$ will be a bad choice if there is a data manifold we care about. If it will be used for generation, such as Experiment 2 in C1.2, $q$ is set to a mixture of $p_\theta$ and $p_\text{data}$. If it will be used only for infering $\log p$ on the data manifold, $p_\text{data}$ will be enough. We all know the NN is not robust on data it hasn’t seen, and so are the marginal networks. Therefore we choose to make choosing $q$ an option in training, such that different $q$ can be chosen for different use cases.
>
> ### **C3: Why Any-Order-ARM is limited under energy-based training setting (in Section 4.3)? (Reviewers sWKU, CcDR, kkqm)**
>
> - This is a great question, we apologize if we didn’t make it super clear. The intuition is that if the probabilities are not consistent over all orderings $\sigma$, for example $\log p_\phi({x} \mid \sigma_1) \neq \log p_\phi({x} \mid \sigma_2)$ for orderings $\sigma_1$ and $\sigma_2$, then $ \log p_\phi ( x ) \neq \mathbb E_\sigma \log p_\phi ( x \mid \sigma )$ even though  $p_\phi(x)=\mathbb E_\sigma p_\phi(x \mid \sigma)$.
>     - For example, let $p(x|\sigma=1) = 2^{-10}$ and $p(x|\sigma=1) = 2^{-1}$, and $p(\sigma=1) = p(\sigma=2)   = \frac{1}{2}$. We can calculate that $p(x) \approx \frac{1}{4}$, but $\mathbb E_\sigma \log_2 p({x} \mid \sigma) = -5.5$, which does not match $\log_2 p(x) \approx -2$.
>     - Due to this reasoning, we cannot take expectation over $\sigma$ for the KL divergence loss $D_\text{KL} = \mathbb E_{p_\phi}[ \log p_\phi(x) - \log p(x) ]$, i.e. $ \mathbb E_{p_\phi} [\mathbb E_\sigma \log p_\phi({x} \mid \sigma) - \log p(x) ] \neq \mathbb E_{p_\phi}[\log p_\phi({x}) - \log p(x) ]$. In other words, it might not match the correct $\log p$.
> - In MLE, this is possible because we are maximizing the expected lower bound: $\log p_\phi({x}) \geq \mathbb E_\sigma \log p_\phi(x \mid \sigma)$.
> - In contrast to ARMs, MaMs match the marginals $\log p_\theta(x)$ to $\log p(x)$ directly and does not need to take expectation for the KL loss term. The second marginal self-consistency penalty term breaks the consistency into small pieces, that allows for taking expectation over orderings.
>
> ### **C4:  Gibbs block sampling with conditionals does not give exact samples from $p_\theta$ and MCMC convergence can be slow. (Reviewers CcDR, kkqm)**
>
> - This is a really good question. The use of persistent Gibbs block sampling is motivated by the classic persistent contrastive divergence (PCD) [1] used for learning energy-based models. The idea is to approximate gradient with samples from persistent MCMC chains and it has shown to be very effective. Here we apply persistent Gibbs block sampling with $p_\phi$ to approximate samples from $p_\theta$. Theoretically, the MCMC chains using $p_\phi$ should converge to $p_\theta$ if self-consistency is enforced.
> - In experiments, we observe that the samples from Gibbs block sampling with conditionals $p_\phi$ converge to the samples from $p_\theta$. On the Ising model problem, we evaluate the $\log p$ of data with $\phi$ and with $\theta$, and compare them using spearman’s and pearson correlation. They are highly correlated on both on-policy $p_\phi$ (using Gibbs sampling) samples (spearman’s  $= 0.9762$, pearson $= 0.9794$) and random samples (spearman’s $= 0.9913$, pearson $= 0.9903$).
> - As mentioned in the paper, importance sampling can be used to make sure we get an unbiased estimate from $p_\theta$. In our experiments, we don’t find this to be necessary since $p_\phi$ and $p_\theta$ improve together due to the self-consistency objective. But it remains to be a good option to have on other problems when $p_\theta$ and $p_\phi$ wildly differ from each other during training.

---

> ### Author Response · Authors · 2023-11-17
> **General response with additional ablation studies/experiments (3)**
>
> ### **C5: Added more baseline and more benchmark (Reviewers 2RBk, sWKU, kkqm)**
>
> - **Baseline**: Per suggestion of reviewer sWKU, we added another baseline MAC [3] that use NN-based mask-tuned ARM to approximately estimate marginals with a sequence of conditionals (**Table 1**) . MAC is slightly better than AO-ARM on test NLL, which is consistent with the findings in [3], since some orderings are easier for learning than others for tasks like images and language. The marginal evaluation time will be exactly the same as AO-ARM since both rely on conditionals. However, MAC cannot perform any-order generation. In comparison, MaM allow much faster marginal $\log p$ inference, and also allow flexible any-order any-block sampling as mentioned in **C1.2**.
> - **Benchmark**: Per suggestion of reviewer kkqm. we tested MaM on CIFAR-10. Due to limited time, we are only able to train MaMs for $200$ epochs ($180$ for conditionals and $20$ for marginals) and achieve a test NLL of $3.34$ bpd (if we continue training to $3000$ epochs, test NLL will get close to $2.69$ bpd shown in the literature [4]). The MaM marginal quality are evaluated using the same procedure on spearman’s and pearson correlation with the AO-ARM model, with $0.9796$ and $0.9774$ respectively. The marginal self-consistency error in eq. (7) is averaged ~0.3 in $\log p$ values. It is conceivable that the same conclusion will apply when training conditionals for more number of epochs and learning marginals from conditionals.
>
> [1] Tijmen Tieleman. Training restricted Boltzmann machines using approximations to the likelihood gradient. In Proceedings of the 25th International Conference on Machine Learning, 2008.
>
> [2] Christopher D. Manning. Stanford CS224N Lectures.
>
> [3] Shih, Andy, Dorsa Sadigh, and Stefano Ermon. "Training and Inference on Any-Order Autoregressive Models the Right Way." arXiv preprint arXiv:2205.13554 (2022).
>
> [4] Emiel Hoogeboom, Alexey A. Gritsenko, Jasmijn Bastings, Ben Poole, Rianne van den Berg,
> and Tim Salimans. Autoregressive diffusion models. In 10th International Conference on
> Learning Representations, 2022

---

### Author Response · Authors · 2023-11-22
**General response -- summary response**

We would like to thank the reviewers once again for the valuable feedbacks and suggestions. In this general response, we briefly summarize our responses that address the concerns and highlight the contributions of our work.

- **Marginalization Self-Consistency with Neural Networks**: Regarding common concerns about self-consistency with NNs, we followed reviewer kkqm's advice and empirically validated self-consistency on a problem with ground-truth PMF, finding close-to-zero errors. In real-world applications, self-consistency errors are found to be non-zero but small. Additional experiments on estimating $\log p$’s, sampling with marginals (with multi-variables), and modeling with scalability (on CIFAR-10) showcase the effectiveness of soft-consistent marginals. We have also discussed in detail the trade-off of exact marginalization (such as PCs) and soft marginalization (such as MaMs).
- Questions regarding **choice of $q$**, **limitations of ARMs** under energy-based training, and **persistent Gibbs block sampling** are answered in details, with experimental and theoretical evidence.
- **Main Contribution - Novelty and Capabilities of Learning Marginals with Neural Networks**:
    - We would like to highlight that the idea of learning marginals with NNs using scalable consistency objectives is *acknowledged to be novel* by all reviewers.
    - In this paper, we demonstrate that *directly modeling marginals with scalability enable new capabilities*, including:
        - Scalable training of any-order models in an energy-based setting, overcoming limitations of ARMs (Section 4.3).
        - Significantly faster marginal inference for comparing $\log p$’s
        - Flexible sampling of multiple variables using marginals

        And soft-consistent marginals are sufficient for accomplishing these capabilities. Designing neural networks that inherently satisfy the self-consistency is a very interesting yet very challenging research problem, however it falls beyond the scope of this paper.


Thank you again for your valuable feedbacks and suggestions that greatly help refine our paper.

---

### Meta-Review · Area_Chair_UBMX · 2023-12-07

**Metareview:**

The paper introduces a density model called Generative Marginalization Models (MaM), which is uses a neural network to assign each sample a marginal density value, where the desired marginal can be selected via a missingness mask as additional input.

The architecture is essentially a variation of AO-ARM, albeit with a different training concept.

The reviews were rather borderline, as the training concept is somewhat incremental, and the experimental results were not very compelling.

While the training concept is theoretically sound, it is rather ad-hoc (enforcing marginal consistency via penalty terms). The experiments are rather limited, probably because it is hard to train MaMs.

**Justification For Why Not Higher Score:**

The reviewers were ultimately not convinced by the ad-hoc training principles and limited experimental evaluation.

**Justification For Why Not Lower Score:**

NA

---

### Decision · Program_Chairs · 2024-01-16

Reject